# Placozoan secretory cell types implicated in feeding, innate immunity and regulation of behavior

Tatiana D. Mayorova[1][¤], Thomas Lund Koch[2], Bechara Kachar[3], Jae Hoon Jung[1], Thomas S. Reese[1], Carolyn L. Smith [4]*

1 Laboratory of Neurobiology, National Institute of Neurological Disorders and Stroke, National Institutes of Health, Bethesda, Maryland, United States of America, 2 Department of Biochemistry, University of Utah, Salt Lake City, Utah, United States of America, 3 Section on Structural Cell Biology, Laboratory of Cell Structure and Dynamics, National Institute on Deafness and Other Communication Disorders, National Institutes of Health, Bethesda, Maryland, United States of America, 4 Light Imaging Facility, National Institute of Neurological Disorders and Stroke, National Institutes of Health, Bethesda, Maryland, United States of America

¤ Current address: Department of Biology, University of Toronto Mississauga, Mississauga, Ontario, Canada
* smithca@ninds.nih.gov

## Abstract

Placozoa are millimeter-sized, flat, irregularly shaped ciliated animals that crawl on surfaces in warm oceans feeding on biofilms, which they digest externally. They stand out from other animals due to their simple body plans. They lack organs, body cavities, muscles and a nervous system and have only seven broadly defined morphological cell types, each with a unique distribution. Analyses of single cell transcriptomes of four species of placozoans revealed greater diversity of secretory cell types than evident from morphological studies, but the locations of many of these new cell types were unknown and it was unclear which morphological cell types they represent. Furthermore, there were contradictions between the conclusions of previous studies and the single cell RNAseq studies. To address these issues, we used mRNA probes for genes encoding secretory products expressed in different metacells in *Trichoplax adhaerens* to localize cells in whole mounts and in dissociated cell cultures, where their morphological features could be visualized and identified. The nature and functions of their secretory granules were further investigated with electron microscopic techniques and by imaging secretion in live animals during feeding episodes. We found that two cell types participate in disintegrating prey, one resembling a lytic cell type in mammals and another combining features of zymogen gland cells and enterocytes. We identified secretory epithelial cells expressing glycoproteins or short peptides implicated in defense. We located seven peptidergic cell types and two types of mucocytes. Our findings reveal mechanisms that placozoans use to feed and protect themselves from pathogens and clues about neuropeptidergic signaling. We compare placozoan secretory cell types with cell types in other animal phyla to gain insight about general evolutionary trends in cell type diversification, as well as pathways leading to the emergence of synapomorphies.

**Data availability statement:** All relevant data are within the paper and its Supporting Information files.

**Funding:** The author(s) received no specific funding for this work.

**Competing interests:** The authors have declared that no competing interests exist.

## Introduction

Placozoans are millimeter-sized, flat, irregularly shaped animals that crawl on surfaces in shallow zones of tropical and subtropical oceans [1–3]. They have no nervous system, muscles or internal digestive cavity. They feed on microalgae and cyanobacteria, which they digest externally in the space between their lower epithelium and the substrate ([4], Fig 1A). They are of interest in an evolutionary context due to their phylogenetic position as sister to the clade that includes Cnidaria and Bilateria[5–7] (Fig 1B) and their simple body plans (Fig 1C) and lifestyles. [8–12] Several dozen distinct placozoan haplotypes have been identified based on sequencing the mitochondrial large ribosomal subunits of specimens collected in different parts of the globe. Based on analyses of nuclear gene sequences of 26 haplotypes, Placozoa is proposed to include 2 classes, 4 orders, 5 families, 8 genera and 26 species/haplotypes [13]. Placozoans primarily propagate by binary fission or budding [14], processes that generate genetically identical clones though sexual reproduction might occur[15].

Although placozoans are genetically diverse, they look alike except for variations in size and shape. Electron microscopic studies of representatives of two proposed orders, *Trichoplax adhaerens* (*TH1), Trichoplax sp.* (*TH2*; order Trichoplacea), and *Hoilungia hongkongensis* (*HH13*; order Hoilungea) showed that they have morphologically identical cell types and body plans [16–20]. The most prevalent cells in the lower epithelium are columnar ventral epithelial cells (VEC), which have an apical cilium surrounded by a collar of microvilli. The cilia are motile and adhere to the substrate during their effective strokes, thereby allowing the animal to crawl on the substrate and to change shape [4,21,22]. Interspersed among the ciliated VEC

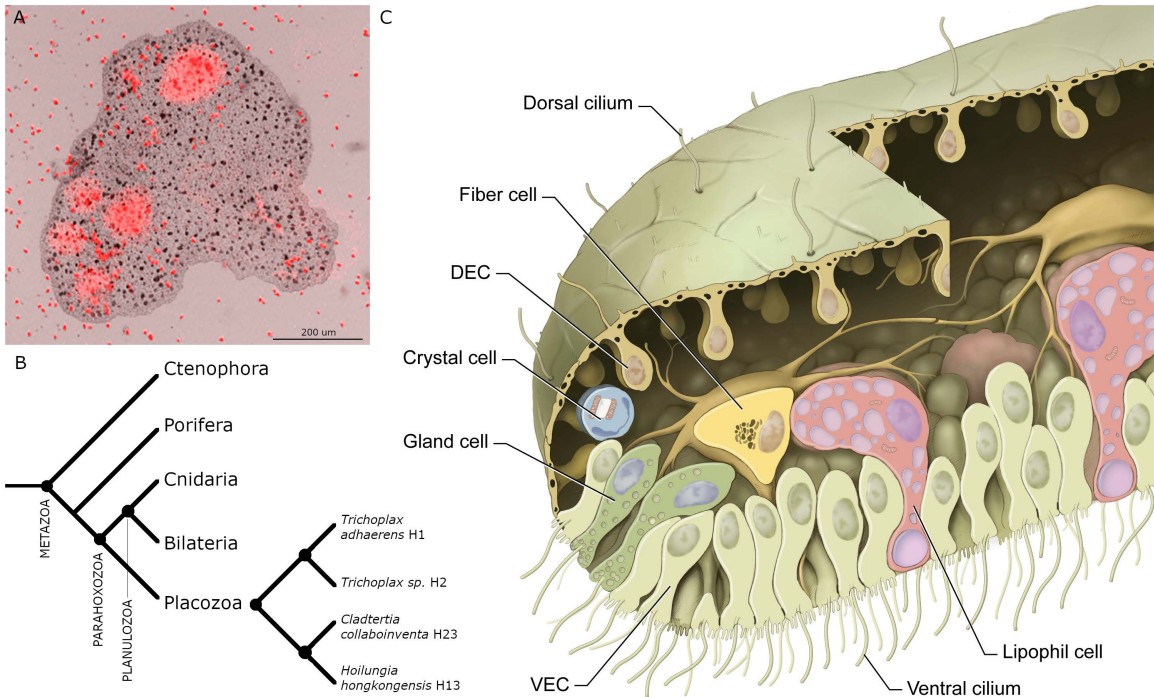

**Fig 1. *Trichoplax adhaerens* (TH1).** (A) Merged brightfield and fluorescence image of *TH1* pausing to feed on *Rhodomonas salina* microalgae (red particles). The image was captured with a confocal microscope (reproduced from [4]). Cells in the ventral epithelium secreted substances that lysed algae, releasing fluorescent phycoerythrin (red clouds). (B) Phylogenetic tree for Metazoa [11] and phylogenetic relationships among four placozoan species here studied. (C) A drawing of main cell types in *TH1* as observed by transmitted electron microscopy (reproduced with permission from [16]). DEC – dorsal epithelial cell; VEC – ventral epithelial cell.

are lipophil cells (LC), so called because they contain multiple large lipophilic secretory granules. When *T. adhaerens* pause to feed on algae, LC secrete a large apical granule whose contents lyses nearby algae [4]. The lower epithelium also contains mucocytes that secrete mucus [18], which the animal requires to crawl on the substrate. The upper epithelium is composed primarily of monociliated cells with broad apical endings that comprise the upper the surface. Their cell bodies are narrower and protrude below. Both dorsal and ventral epithelia contain several morphologically distinct types of gland cells that possess a cilium and may be sensory [18,22,23]. The space between the upper and lower epithelial layers is occupied by a layer of fiber cells, which have long branching processes that contact each other as well as the other cell types. Fiber cells phagocytose bacteria and cellular debris and engage in wound healing [24], functions associated with macrophage-like cells in other animals. Inside the rim of the animal, where dorsal and ventral epithelia meet, are regularly spaced crystal cells that appear to be functional statocysts [25].

Bioinformatic analyses of *TH1* [5,6,26–29] reveal a rich repertoire of secretory proteins including digestive enzymes, proteins implicated in innate immunity in other animals, and precursors of peptides, some of which elicit changes in the behavior of the animals [22,23]. Single cell RNA sequencing (RNA-Seq) analyses of *TH1* and three other species of Placozoa [6,30] revealed that many of these secretory products are differentially expressed. Cells expressing several digestive enzymes have been localized by fluorescence *in situ* hybridization (FISH) [6,18] and some peptidergic cell types have been localized by immunolabeling [18,22,23]. However, the cells that express many secretory products remained unidentified.

In the present study, we used FISH probes for genes encoding secretory products that are specifically expressed in *TH1* metacells classified as lipophil, gland, or epithelial and a subset of the peptidergic metacells to localize the cells in whole mounts and in dissociated cell cultures and by staining them with fluorescent dyes or immunolabels. The compositions of the secretory granules in LC, VEC and dorsal epithelial cells were further investigated with electron microscopic techniques. We used fluorescent dyes and light microscopy to visualize secretory processes in live animals feeding on algae.

Our findings yield a more detailed picture of the placozoan body plan than was apparent from previous morphological studies and provide insight into the mechanisms that the animals use to feed and maintain homeostasis. Comparative analyses of the secretomes of metacells in *TH1*,*TH2*, *HH13*, and *Cladtertia collaboinventa* (*CH23)* revealed numerous similarities and several intriguing differences suggesting important differences across placozoan species.

## Results

### Main secretory cell types in the ventral epithelium

*Trichoplax adhaerens H1* lipophil cell (LC) metacell clusters highly express multiple genes that are not expressed or expressed at much lower levels in other metacells (S1 Fig, S3 Fig) identified by single cell RNA-Seq studies [6,30] We obtained fluorescence *in situ* hybridization (FISH) probes for three of them: *Ta 58643* (*Ta Tetraspanin*); *Ta 29105* (*Ta GABA transporter*); *Ta 63996* (*TH1* secretory protein). We used probes for *TH1*orthologs of the precursors of *trypsin (Ta 63128)*, *chymotrypsin* (Ta 63088) and *secretory phospholipase A2* (*sPLA2; Ta 57870*) to label the cell types classified as "digestive gland cells" in the RNA-Seq studies[6,31].

Cells (presumably LC) co-labelled with probes for *Ta tetraspanin*, *Ta GABA transporter* and *Ta 63996* were present throughout the central region of the ventral epithelium but absent in a zone <10 μm from the rim (Fig 2A and S2A Fig). Cells labeled with probes for digestive enzymes (*Ta trypsin*, *Ta chymotrypsin*) were interspersed with LC in the central part of the animal but were absent <60 μm from the rim (Fig 2B and S2B Fig). The label in these cells

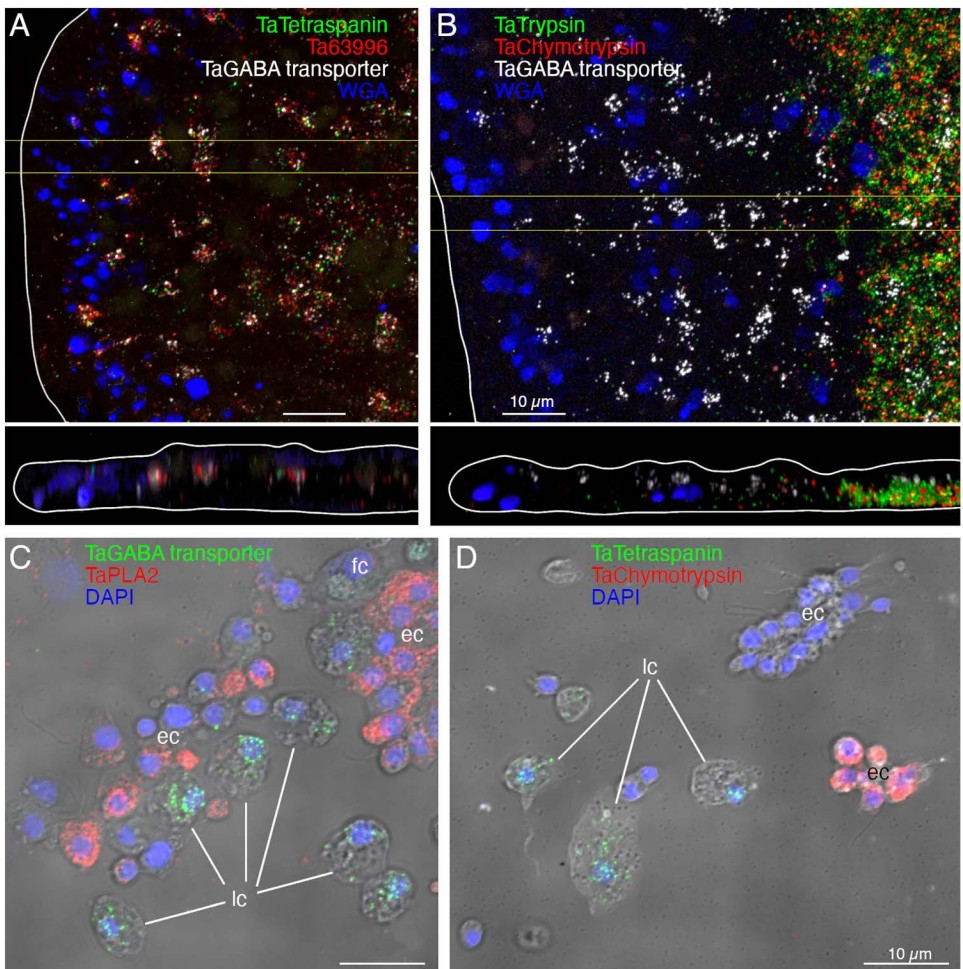

**Fig 2. Fluorescence *in situ* hybridization (FISH) localization of expression of lipophil and digestive gland cells markers in *T. adhaerens* H1 wholemounts (A, B) and dissociated cell preparations (C, D).** Images of wholemounts are horizontal (xy) and vertical (xz, from boxed region on xy) maximum intensity projections encompassing ~1/4 of the diameter of the animal (edge of the animal is outlined white). Mucocytes are labeled with WGA. (A) Lipophil cell markers (*Ta Tetraspanin*, *Ta* 63996, and *Ta GABA transporter*) are co-expressed in scattered clusters ~8 μm in diameter throughout the central region of the animal, starting ~ 10 μm from the rim. (B) Digestive cell markers (*Ta Trypsin* and *Ta Chymotrypsin*) are highly expressed in a region starting ~60 μm from the rim. Cells expressing a lipophil specific marker (*Ta GABA transporter*) are interspersed among the digestive gland cells in this region. (C and D) Lipophil cell markers (C, *Ta GABA transporter*, and D, *Ta Tetraspanin*) and digestive gland cell markers (C, *Ta PLA2*, and D, *Ta Chymotrypsin*) are expressed in different populations of cells. Nuclei are labelled with DAPI (maximum intensity projections merged with DIC). Color separated images corresponding to A–D are shown in S2 Fig. fc – fiber cells; ec – epithelial cells; lc – lipophil cells. Scale bars 10 μm.

was closer to the ventral surface (Fig 2B, bottom and S2B Fig) than the label in LC (Fig 2A, B, and S2A, B Fig). Dissociated cells expressing LC specific markers were larger (~8 μm) than cells expressing digestive enzymes (<5 μm; Fig 2C, D, and S2C, D Figs) and contained large granules, a hallmark of LC. Unlabeled ventral epithelial cells, identified by their small sizes and cylindrical shapes, also were present (Fig 2D). We previously reported that LC express digestive enzymes based on sequential labeling with vital dyes in live dissociated cultures and FISH probes for precursors of digestive enzymes after fixation of the cells [18]; in this study, we found that the expression level of digestive enzymes by LC was very low, as reported [6].

Lipophil cells in thin section from animals prepared by high pressure freezing and freeze substitution contained granules of varying sizes and appearances (Fig 3A). Granules in the basal part of the cell, the location of the Golgi complex where the granules originate [4], ranged in size from a few hundred nm to 2 μm and were electron lucent (the same as background) (Fig 3A, B). Some granules contained membrane bound profiles with content that resembled cytoplasm (Fig 3B). Analysis of serial thin sections (Fig 3B), freeze fracture replicas (Fig 3C), and tomograms (Fig 3D, E) of these granules showed that the profiles represented cytoplasmic protrusions into the granule. Granules closer to the apical pole of the cell were up to 5 μm in diameter and contained variable amounts of electron dense material as well as areas that were electron lucent (Fig 3A, B). The largest granule resided close to the apical surface of the cell. These apical granules often had a ring of electron dense material under their

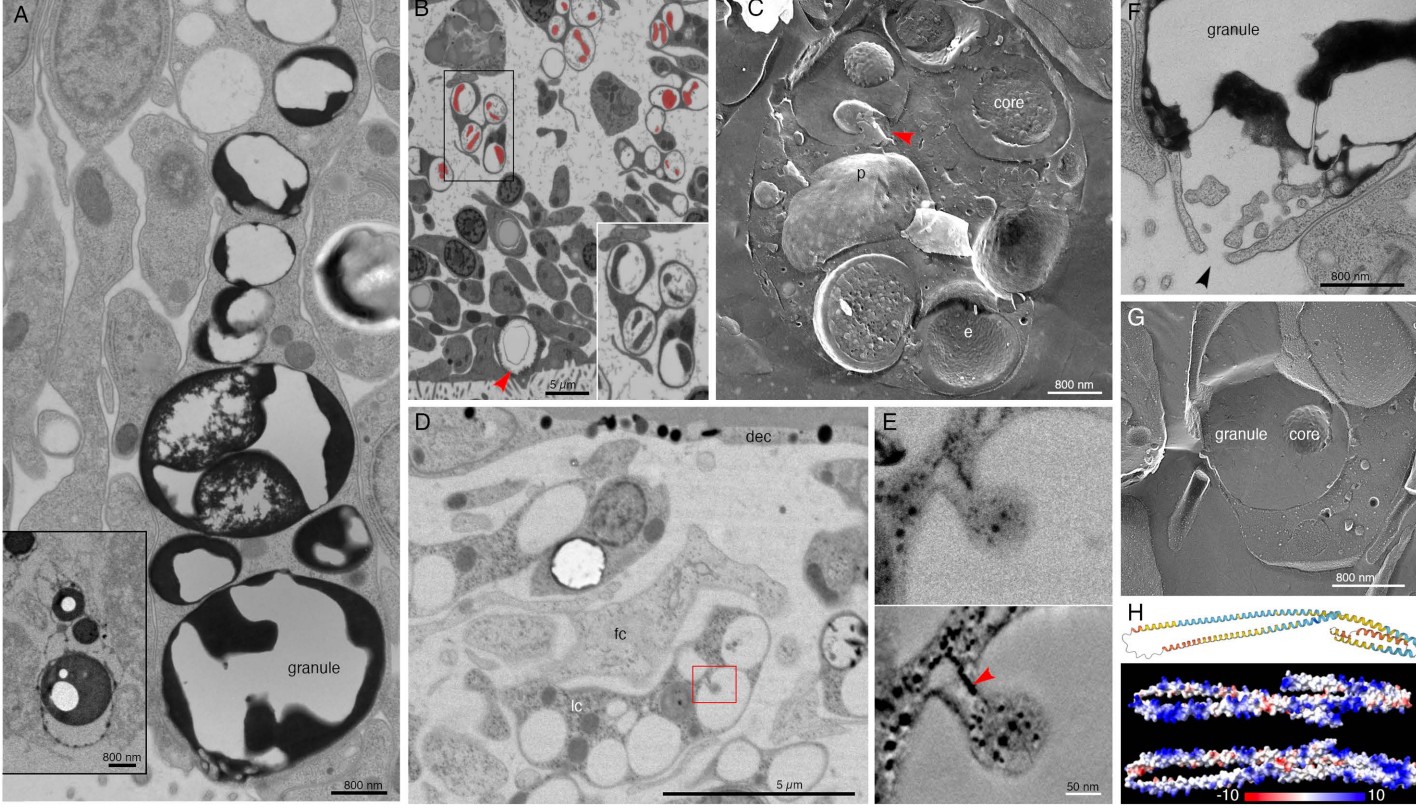

**Fig 3. Ultrastructural features of lipophil cells and the structure of a highly expressed lipophil cell secretory protein.** (A) Transmission electron microscopy (TEM) image of a thin section from a lipophil cell (LC) with osmiophilic material in its granules. The inset shows a TEM image of a section from an animal fixed with osmium, which better preserved the lipidic content of the granules and their cores. (B) SEM image of a section taken in a backscatter mode shows that granules in basal portions of LC have a dense core (artificially colored red) surrounded by electron lucent content. Inset is an enlarged view of the boxed region. Arrowhead indicates an apical LC granule. (C) Freeze fracture replica showing a basal part of a LC with multiple granules. Note dual component nature of the granules. A fracture through the interior of a core reveals heterogeneous content resembling cytoplasm. Arrowhead marks a protrusion emanating inward from the granule wall. (D) TEM image of 100 nm thick section showing basal region of a LC as inferred from its proximity to the dorsal side of the animal. Red box depicts a region with a protrusion inside a granule, further studied by EM tomography in (E). (E) Upper panel is an EM projection showing direct connection between the protrusion and the inner surface of the granule. Lower panel is a 1.75 nm thick virtual slice through a reconstructed volume of the tomogram. Note that ER (arrowhead) penetrates the protrusion. (F) TEM image of an apical part of an LC showing exocytosis (arrowhead) of a large granule. Note osmiophilic material and membranous particles located near the site of exocytosis. (G) Freeze fracture replica showing an apical part of a LC and its apical granule. (H) A protein present exclusively in LC secretome is largely composed of alpha helices (AlphaFold per-residue confidence score, pLDDT, color-coded: dark blue > 90 very high confidence; light blue 90 > pLDDT > 70 confident; yellow 70 > pLDDT > 50 low confidence; and orange < 50 very low confidence) and is positively charged (electrostatic map created with ChimeraX). dec – dorsal epithelial cell; **e** – e-face; fc – fiber cell; lc – lipophil cell; **p** – p-face. Scale bars 5 μm (B, D), 800 nm (A, C, F, G), and 50 nm (E).

membrane (Fig 3A, F), but some were completely devoid of electron dense material. Small (<100 nm) membrane bound inclusions typically were present near the portion of the granule membrane that was closest to the apical surface of the cell (Fig 3A, F). Examination of these small inclusions in serial sections confirmed that they resided in the interior of the granule. We saw a single example of a LC granule with an open fusion pore and the content of the granule, including small membrane bound inclusions, exposed to the exterior (Fig 3F). The continuity of the granule and plasma membranes at the edge of the pore leads us to believe that this profile represents a stage in exocytosis of a LC granule.

The variable appearance and patchy distribution of the content of LC granules suggested that the content was partially extracted by the procedure used for freeze substitution and fixation. The apical granule in LC in thin sections from animals fixed in glutaraldehyde and osmium simultaneously in aqueous solution at room temperature was a large electron dense sphere with one or two electron lucent inclusions inside (clearer than background plastic; Fig 3A, inset). The electron dense material may represent unsaturated lipids, which are osmiophilic [31] and might be extracted by the organic solvents used for freeze-substitution of frozen specimens. The clear inclusions contain substances that do not bind the uranyl acetate and lead citrate grid stains suggesting that they are not proteinaceous. Analysis of freeze fracture replicas from rapidly frozen animals provided further evidence of the heterogeneous nature of the content of LC granules (Fig 3C, G). A fracture face through the outer content of the granule was smooth in appearance while the fracture face of the granule core was rough.

The uncharacterized protein *Ta 63996* that served as one of our markers for LC is the second most highly expressed genes in *TH1* LC metacells (S1 and S3 Figs) and the identical gene in *TH2* (*006628*) is the most highly expressed gene in *TH2* LC metacells (S4 Fig). The gene was annotated as laminin subunit *beta,* but it lacks a laminin domain and laminin-type EGF domains. It has only weak homology with a region located downstream of the critical laminin domains in laminin beta subunit orthologs in other animals and is much shorter (346 aa versus ~1700 aa) than in other species. The gene has a signal peptide and no trans-membrane domains. Only LC express this putative secretory protein in *TH1* although some cells classified as "upper-epithelial-like" express it in *TH2*. The protein structure predicted by AlphaFold includes extended helical regions with abundant basic amino acids (Fig 3H). Many lysines are grouped in pairs, composing a KK motif, a well-known cleavage site in neuropeptide prohormones. However, this motif is not conserved beyond Trichoplacidae, and LC do not express any known prohormone convertases or cathepsins. We found no other highly expressed secretory proteins that lack transmembrane domains in *Trichoplax* LC metacells.

Two LC metacell clusters, LC1 and LC2, were identified in *HH13* (S5 Fig) and *CH23* (S6 Fig). The *HH13* LC2 metacell cluster expressed a gene whose sequence was 34% identical and 52% positive to the uncharacterized secretory protein *Ta 63996*. In *CH23*, both LC1 and LC2 metacells expressed an ortholog (e$^{-45}$) of *Ta 63996*. All LC metacells in *TH1, TH2, HH13* and *CH23* contained a gene identical to the Tetraspanin (*Ta 58643*) we used as a second marker for LC. The third marker we used for LC, *Ta 20105,* was annotated as a solute carrier (*GABA transporter*). It was expressed in all metacells classified as LC in *TH1* and *TH2* but only in LC1 metacells in *HH13* and *CH23*. All LC metacell clusters in *TH1, TH2, HH13* and *CH23* highly expressed several fatty-acid binding proteins, V-type ATPases and solute carriers, and the lysosomal membrane glycoprotein, LAMP2 (S3–S6 Figs).

We further investigated the morphology of cells that express digestive enzymes by co-labeling dissociated cell preparations with FISH probes for the precursors of trypsin and sPLA2 and an antibody against acetylated tubulin, a component of cilia. Cells co-expressing *Ta trypsin* and *Ta sPLA2* often occurred in clusters (Fig 4A). The cells were small and cylindrical in shape and many of them possessed a cilium. Labeled cells lacking a cilium may

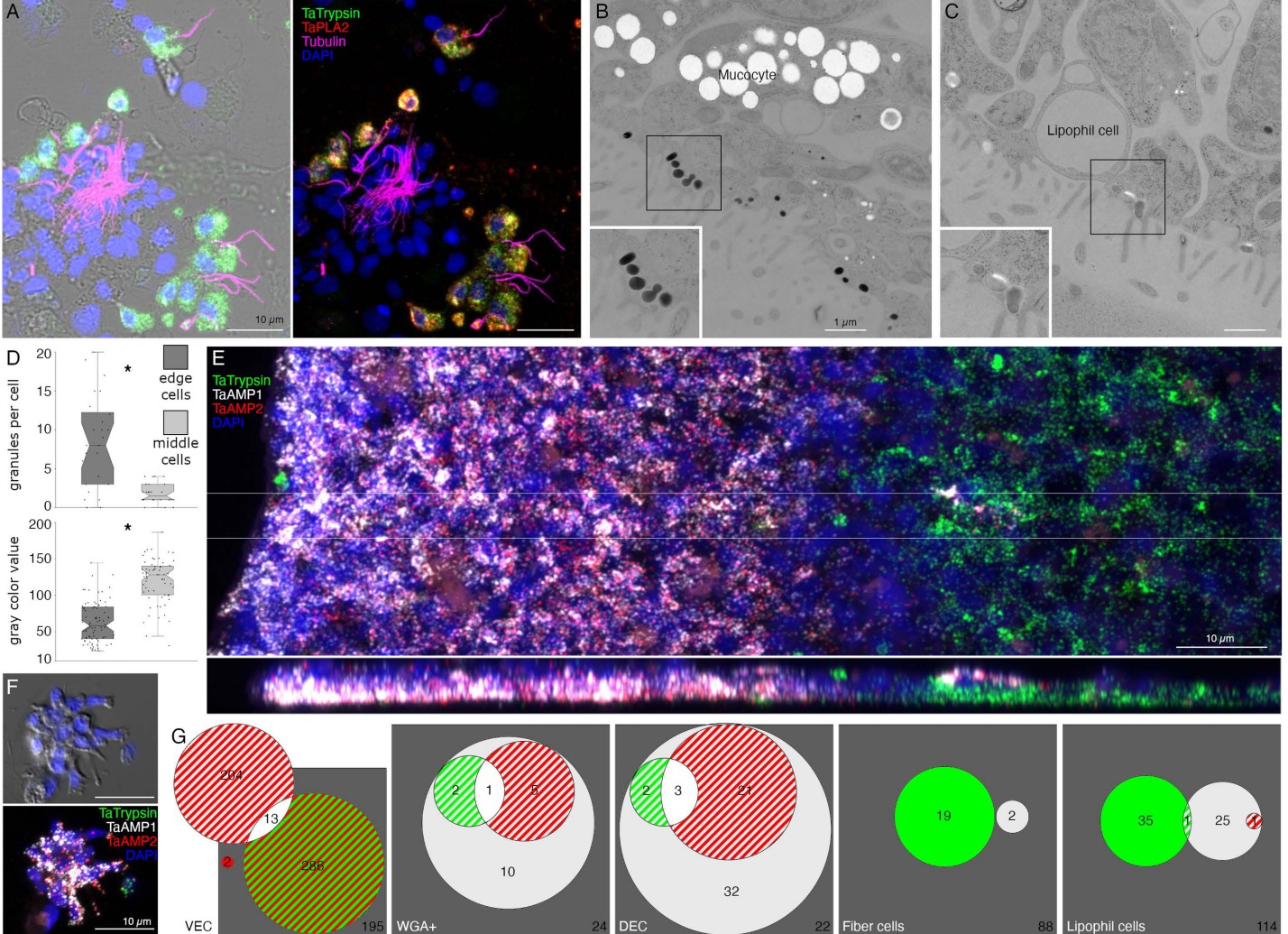

**Fig 4. Two monociliated ventral epithelial cell (VEC) types: digestive gland cells and cells expressing putative antimicrobial peptides (AMPs).** (A) Cells secreting digestive enzymes bear a cilium (combination of FISH for *Ta Trypsin* and *Ta PLA2* and immunolabelling for tubulin; left panel shows fluorescence merged with DIC). (B, C) TEM shows that both peripheral (B) and central (C) ciliated epithelial cells have ~500 nm diameter granules located close to the apical surface. Boxed regions are magnified in insets. (D) The granules in peripheral cells are more electron dense and abundant than those in cells in more central regions. * p<0.05. (E) Trypsin is highly expressed in the central region of the ventral epithelium, while *AMP1* expression is restricted to the peripheral region. (F) In dissociated cell preparations, two distinct subpopulations of cells expressing either *AMP1* or *Ta Trypsin* are apparent. Most *AMP1* expressing cells strongly express *AMP2* whereas *Ta Trypsin* expressing cells show weak *AMP2* expression (not visible in E). A dissociated trypsin+ cell (F, green) contains one red *AMP2* grain. (G) Venn diagrams based on cell counts show that three quarters of VEC express either *AMP1* or *Ta Trypsin* along with *AMP2*. Some mucocytes and DEC express *AMP1* with or without either *AMP2* or *Ta Trypsin*. Only a few lipophil and fiber cells express *AMP1* or *Ta Trypsin*. Color coding for Venn diagrams is the same as the fluorescence colors on E and F; double co-expression is indicated as strips of respective colors, triple co-expression is white, and the absence of expression is dark gray. Numbers of counted cells are shown. Scale bars: 10 μm (A, E, F) and 1 μm (B, C).

have lost their cilium during dissociation since cilia that were not attached to a cell also were present. Ciliated ventral epithelial cells (VEC) are the most prevalent cell type in the ventral epithelium. Given the high density of cells expressing digestive enzymes in the central part of the ventral epithelium and their possession of a cilium, we identify "digestive gland cells" as a subtype of VEC (S1 and S3–S6 Figs).

As VEC in the peripheral part of the ventral epithelium (pVEC; within ~60 μm of the rim) do not express digestive enzymes we speculated that their ultrastructure might differ from that of the centrally located VEC (cVEC) that express genes encoding precursors of digestive

enzymes. Comparison of pVEC and cVEC within the same thin sections revealed differences in their secretory granules (Fig 4B–D). Peripheral VEC, identified based on their proximity to the rim, had numerous electron dense secretory granules near their apical surfaces (Fig 4B, D). Ventral epithelial cells located in the central region, in the vicinity of LC, had fewer granules ($p = 1.96E-05$) and the granules were less electron dense ($p = 1.26E-17$) (Fig 4C, D).

Metacells identified as "epithelial" in $TH1$ [30] or "lower epithelial" [6] expressed multiple genes that were predicted precursors of secretory proteins/peptides (InterPro). We noticed that the sequences of five of them resembled those of arminins, a type of antimicrobial peptide (AMP) expressed in *Hydra* [32], in that they were ~150–250 aa in length, had a signal peptide, a highly acidic N-terminal region and alkali C-terminal region that includes aliphatic amino acids and DE cleavage motifs (S7 Fig) common in placozoan and cnidarian prepropeptides [33,34]. The N-terminal domain of one of them, *Ta 55945*, was annotated "alpha defensin" (Interpro). The signal peptide and acidic N-terminal regions of arminin prepropeptides are removed to produce active arminin peptides. Arminin peptides have a C-terminal glycine and are thought to be amidated. The sequences of the *Trichoplax* arminin-like peptides have a C-terminal glycine, as is required for amidation by peptidylglycine α-amidating monooxygenase (PAM). An AMP analysis tool (Antimicrobial Peptide Data, Univ. Nebraska Medical Center) identified hydrophobic and basic regions in the C-terminal part of the Ta AMP sequences that might interact with membranes.

We obtained FISH probes for two of the arminin-like genes, *Ta 55945* and *Ta 56030*, and will refer to them as *AMP1* and *AMP2*, respectively. Both probes strongly labeled cells in the peripheral part of the ventral epithelium (Fig 4E). The AMP expressing cells were densely packed and showed little overlap with cells expressing digestive enzymes. Scattered cells in the dorsal epithelium showed moderate expression of *AMP1* or both *AMP1* and *AMP2*. Most of the AMP-expressing cells in dissociated cell preparations had morphological features typical of VEC: they were small, cylindrical in shape and possessed a cilium (Fig 4F). They often occurred in clusters with other AMP-expressing cells. Most *AMP1* expressing VEC strongly expressed *AMP2*. *Trypsin*-expressing VEC weakly expressed *AMP2*, containing fewer and smaller fluorescent grains than cells co-expressing *AMP1* and *AMP2*. Some WGA-positive mucocytes and DEC expressed *AMP1* with or without *AMP2*. A small fraction of LC and fiber cells contained one or a few *AMP1* or *Ta trypsin* positive grains (Fig 4G) and may represent cells transdifferentiating from cVEC to lipophil or fiber cells, respectively [6].

The genes encoding *AMP1*, *AMP2*, and a third arminin-like prepropeptide (*Ta 60631; AMP3*) are the three most highly expressed genes in *Trichoplax* H1 classified as "lower epithelial" metacells [6]. Our findings demonstrate that these metacells correspond to peripheral VEC (pVEC; S1 and S3 Figs). The *TH2* metacells classified as "lower epithelial" contain genes that are nearly identical to *AMP1*, *AMP2* and *AMP3* (S4 Fig). The *AMP1* and *AMP3* genes are the most highly expressed genes these metacells. *Hoilungia H13* and *CH23* metacells classified as "lower epithelial" contain a gene that is more than 55% identical and 70% similar to *AMP3* and a gene that is more than 40% identical and 60% similar to *AMP2* (S5 and S6 Figs). No *AMP1* ortholog was found in *HH13* or *CH23*.

## Main secretory cell types in the dorsal epithelium

The most prevalent cells in the dorsal epithelium are monociliated dorsal epithelial cells (DEC) [16]. Their broad (~10 μm) polygonal-shaped apices pave the dorsal surface. Their cell bodies are narrower and extend into the interior, where they are surrounded by processes of fiber cells. Freeze fracture replicas of the apices of DEC revealed exoplasmic (e) faces and protoplasmic (p) faces of numerous, large (500 nm) secretory granules (Fig 5A). Electron

micrographs of transverse thin sections through the apices of DEC showed dark secretory granules near their dorsal surfaces (Fig 5B). The dark granules bound the lectin wheat-germ agglutinin (WGA) conjugated to nanogold (Fig 5B, inset). WGA also labeled the outer surfaces of DEC. At the rim, cells with morphological characteristics of DEC were adjacent to cells with morphological characteristics of VEC (transition zone). Secretory granules in pVEC were grey or dark like those in DEC but bound much less WGA label (Fig 5C, insets).

Many DEC in wholemounts and dissociated cell preparations, identified based on their content of granules stained with WGA conjugated to a fluorescent dye, were labeled by a FISH probe for *Ta 60661* (Fig 5D, E), a gene that is highly expressed in *TH1* metacells classified as "epithelial" or "upper epithelial" [6,30]. Labeled cells were present throughout the dorsal epithelium except in a region within ~15 µm of the rim (Fig 5D, E). Their distribution was patchy, suggesting that only a subset of DEC expressed this gene.

*Trichoplax H1* gene *Ta 60661* is identical to a gene discovered in a search for orthologs of genes implicated in innate immunity in *TH2* [35]. The gene was identified as belonging to a class of secreted glycoproteins called "intelectins" based on its content of a fibrinogen-related domain (FReD). Intelectins have been implicated in defense against bacteria in both vertebrates and invertebrates [36]. Ten of thirty-one intelectin-like genes identified in *TH2* are uniquely expressed in *TH1* and *TH2* metacells classified as "upper epithelial" (S1, S3 Figs, Fig 4). These metacells also contain a gene annotated "membrane-associated mucin 4-like glycoprotein" based on the presence of von Willebrand factor type D (VWD), Scavenger Receptor Cysteine-Rich (SRCR) and multiple EGF-like domains. *Hoilungia H13* metacells classified as "upper epithelial" include genes encoding five of the intelectins and the mucin-4 gene, but

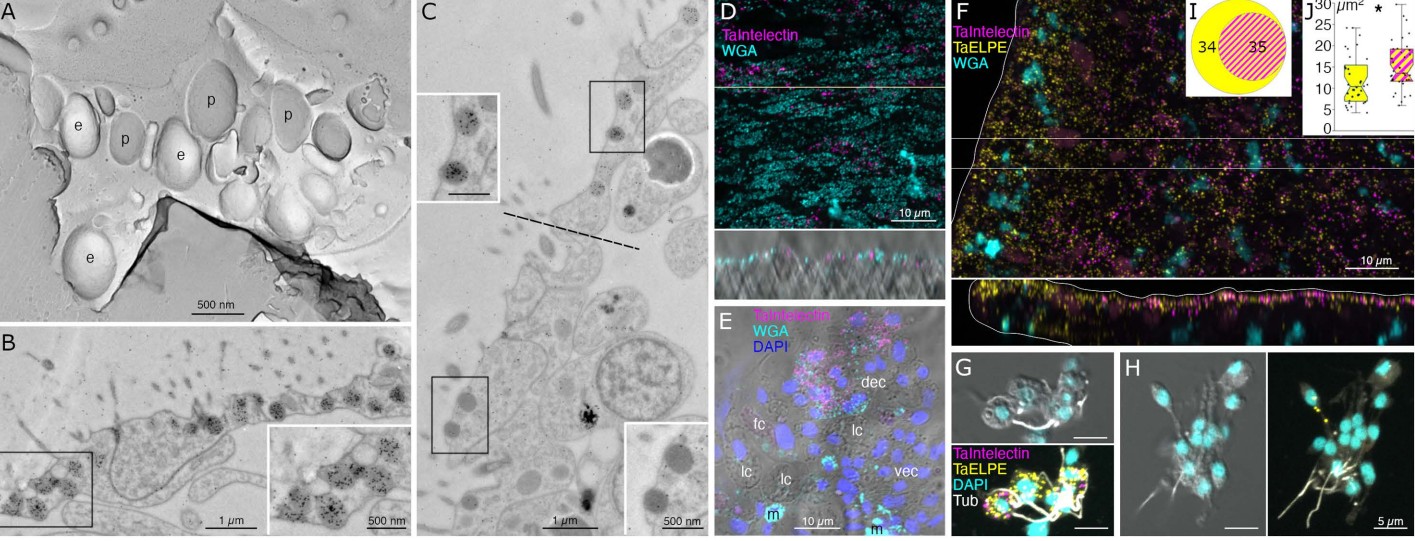

**Fig 5. Main secretory cells in the dorsal epithelium.** (A) Freeze fracture replica at the apex of a dorsal epithelial cell (DEC) imaged in TEM reveals e- or p-faces of numerous ~500 nm diameter secretory granules. (B, C) TEM of ultrathin sections labelled with nanogold conjugated WGA. Transverse section in the dorsal epithelium (B) shows multiple WGA-stained granules in ciliated DECs; insets show enlarged view of boxed regions. Transverse section at the transition region between dorsal and ventral epithelia (C, the border is demarcated by dotted line) shows that DEC granules bind more WGA than do morphologically similar granules in VEC. (D–H) Confocal images of wholemounts (D, F) and dissociated cell preparations (E, G, H). Many DEC express *Ta Intelectin 60661* (D, whole mount; E, dissociated cells) as evident from co-labeling with WGA. Mucocytes (m) label intensely with WGA, but do not express Ta *Intelectin 60661*. Other cell types (VEC, lipophil (lc) and fiber (fc) cells) are not labeled. (F) *Ta ELPE* is expressed in nearly all DEC and a few VEC (see xz inset in F and color separated images in S9A Fig); (G, I) About half of *Ta ELPE*+ co-expresses *Ta Intelectin 60661*. (H) Some VEC, identified based on their small sizes and cylindrical shapes express *Ta ELPE* but not *Ta Intelectin 60661*. (J) Cells co-expressing *Ta ELPE* and *Ta Intelectin 60661* are larger than those expressing only *Ta ELPE*. **e** – e face; fc – fiber cells; lc – lipophil cells; **m** – mucocytes; **p** – p face.

not intelectins *Ta 60661* and *Ta 61411* (S5 Fig). Instead, these genes are expressed in a metacell classified as "gland". *Cladtertia H23* metacells classified as "upper epithelial" express four of the intelectins expressed in upper epithelial metacells in *TH1*, *TH2* and *HH13* (S6 Fig).

The metacells representing DEC also express a prepropeptide identified in *TH1* genomes and transcriptomes that is predicted to produce peptides with C-terminal amino acids Glu-Leu-Pro-Glu, or ELPE [6,23,33]. We will refer to this gene as *Ta ELPE*. The names, sequences and predicted mature peptides produced by this and other prepropeptide genes investigated in this study are listed in S8 Text.

A FISH probe for *Ta ELPE* prepropeptide labeled cells throughout the dorsal epithelium including those <15 μm of the rim (Figs 5F and 6A1 and S9A Fig). Approximately half of the *Ta ELPE*-positive DEC co-expressed *Ta Intelectin 60661* (Fig 5G–I). A few VEC, identified based on their small sizes and cylindrical shapes, expressed *Ta ELPE* but not intelectin *Ta Intelectin 60661* (Fig 5F, H). The mean area of cells expressing both *Ta Intelectin 60661* and *Ta ELPE* was greater (p=0.01) than that of cells expressing *Ta Intelectin 60661* without *Ta ELPE* (Fig 5G, H, J) because the latter group included the *Ta ELPE*-positive VEC, which were smaller than DEC.

Scattered cells in the dorsal epithelium and a few cells in the ventral epithelium located 10–40 μm from the rim were labeled with a probe for a gene encoding a secretory protein with no known functional domains, *Ta 63786* (Fig 6A1). The labeled cells did not express *Ta ELPE* (Fig 6A1 inset and A2). The *Ta 63786+* cells in the ventral epithelium, but not those in the dorsal epithelium, co-expressed a gene *Ta 64402* encoding a secretory peptide that bears the same *Hydra* arminin-like features as *AMPs 1*, *2* and *3*, including a signal for C-terminal amidation, which we call *AMP4* (Fig 7E, F). Dissociated *Ta 63786+* cells had a cilium but were smaller and narrower (p=0.004) than *Ta ELPE+* and *Ta Intelectin 60661+* DEC cells (Fig 6A2, A3). The metacells that express *Ta 63786* and *AMP4* were classified as "peptidergic" or "epithelial unknown" [6,30]. These metacells also contain a gene encoding an intelectin (*Ta 64393*) that is not expressed in any other metacell (S1 and S3 Figs). *Trichoplax* H2 metacells classified as "gland", or "epithelial lower" [6], express genes identical to *Ta 63786* and *Ta 64393*, and a gene that is nearly identical to *AMP4* (S4 Fig). The DEC labeled by the probe for *Ta 63786*, but not the probe for *AMP4* likely correspond to *TH1* metacells C102-107 (S3 Fig) and *TH2* metacells C133 and 134 (S4 Fig), which highly express *Ta 63786* but do not express *AMP4* or the intelectin. Metacells *TH1* C103 and *TH2* C133 and 134 were reported to co-express *Ta 63786* and *Ta ELPE* [6] but we did not observe cells co-expressing these genes in our microscopy studies. *Hoilungia H13* metacells C177-183 express orthologs of *Ta 63786* and *Ta 64393* intelectin (S5 Fig), but no *AMP4* ortholog was found in this species. *Cladtertia H23* lacks orthologs of *Ta 63786*, *Ta 64393* intelectin and *AMP4*. No ortholog of *Ta ELPE* was found in *HH13* or *CH23*.

## Morphology and locations of peptidergic secretory cells

A probe for the precursor of Leu-Phe (*Ta LF*) peptides labeled a row of cells in the dorsal epithelium 5–10 μm from the rim and more weakly labeled scattered cells in the ventral epithelium >40 μm from the rim (Fig 6B1). A probe for a secreted astacin-like metalloendopeptidase, *Ta 26557*, intensely labeled the *Ta LF+* cells in the ventral epithelium but not the *Ta LF+* cells in the dorsal epithelium (Figs 6B1-4 and 7A). Both *Ta LF+/Ta astacin+* and *Ta LF+/Ta astacin-* cells were cylindrical in shape, possessed a cilium (Figs 6B2, B3, and 7B), and were similar in size (Figs 6B5 and 7C, D). The positions of *Ta LF+/Ta astacin+* cells are like those of cells classified as ventral Type 3 gland cells in ultrastructural studies of *Trichoplax* [18]. Type 3 gland cells contain small granules with textured content. The *Ta LF+/Ta astacin+* cells were intermingled with cells expressing *Ta sPLA2*, but only a few cells co-expressed *Ta astacin* and

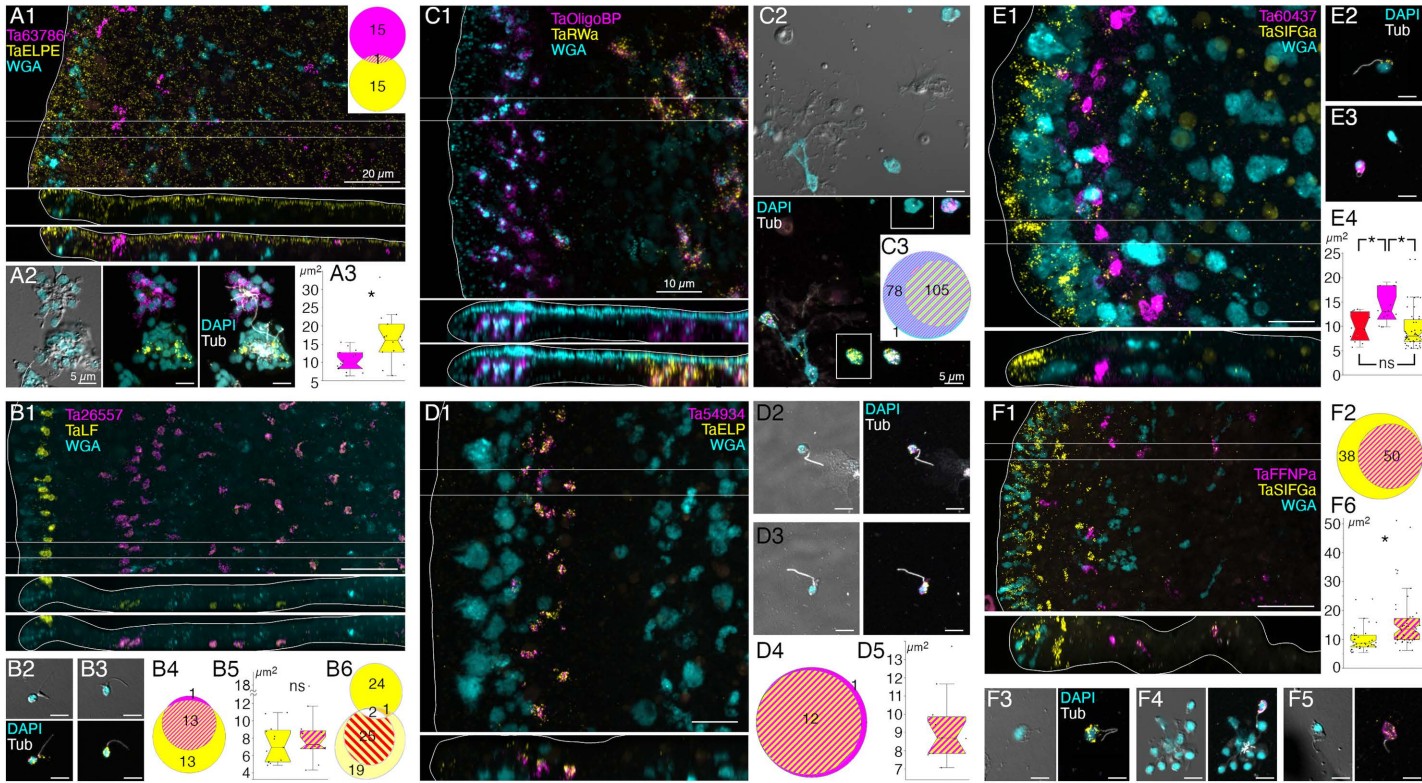

**Fig 6. Localization and characterization of peptidergic secretory cells.** *Trichoplax* wholemounts and dissociated cell preparations were labeled with FISH probes for different prepropeptides and other cell-type specific proteins. Mucocytes were labeled with fluorescent-conjugated WGA in wholemounts. Cilia were immunolabeled with an antibody against acetylated tubulin and nuclei were labeled with DAPI in dissociated cell preparations. Circle plots and box plots use the same color coding as fluorescence images; co-expression is indicated as stripes of respective colors. (A) Prepropeptide *Ta ELPE* (A1) was expressed by DEC throughout the animal and by some VEC near the rim. Interspersed among the *Ta ELPE+* DEC were scattered cells that expressed an uncharacterized protein (*Ta 63786*). Probes for *Ta ELPE* and *Ta 63786* labeled distinct populations of ciliated cells in dissociated cell preparations (A1, circle plot; A2). *Ta ELPE+* cells were larger in area than *Ta 63786+* cells (A3). (B) *Ta LF* prepropeptide (B1) was strongly expressed in a row of cells in the dorsal epithelium 5 - 10 μm from the rim and more weakly expressed in scattered cells in the ventral epithelium in a region starting 40 μm from the rim. The yellow channel in the color-merged xy and xz images were dislayed with gamma= 0.54 to enhance the visibility of the weakly labeled cells. The yellow channel is displayed with gamma=1.0 (linear) in the upper xz image. The *Ta LF+* ventral epithelial cells co-expressed an astacin-like metalloendopeptidase (*Ta 26557*; B1, B2), while dorsal *Ta LF+* cells did not (B1, B3, B4). Both *Ta LF+/Ta 26557+* cells (B2) and *LF+/Ta 26557-* cells (B3) were small ciliated cells (B5). Circle plot B6 shows that *Ta LF+* cells (bright yellow) were distinct from *Ta ELPE+* cells (light yellow) and *Ta Intelectin 60661+* cells (red stripes). (C) Mucocytes were labeled with fluorescent WGA and a probe for a secreted oligosaccharide binding protein (*Ta OligoBP; 63702*; C1). The central population of mucocytes co-expressed *Ta RWa* prepropeptide (C1-3). Both *Ta RWa-* and *Ta RWa+* mucocytes lacked a cilium (C2, tubulin label absent). The rectangular insets show *Ta RWa-* (top) and *Ta RWa+* mucocytes (bottom) without the magenta (*Ta OligoBP*) fluorescence channel. (D) *Ta ELP* prepropeptide and an astacin-like metalloendopeptidase (*Ta 54934*; D1) were co-expressed in a row of cells in the ventral epithelium 15 to 30 μm from the rim. The labeled cells were ciliated (D2, D3), co-expressed both genes (D1-4) and had an area of about 9 μm² (D5). (E) Probes for *Ta SIFGa* prepropeptide and for a second uncharacterized secretory protein (*Ta 60437*) labeled separate populations of cells: *Ta SIFGa+* cells were prevalent in the dorsal epithelium of the rim (E1) and more sparsely distributed in the dorsal and ventral epithelium further in the interior; *Ta 60437+* cells were in a row 20 to 30 μm from the rim (E1). Both the *Ta SIFGa+* and *Ta 60437+* cells were ciliated (E2, E3). The *Ta 60437+* cells were larger than the *Ta SIFGa+* cells (E4,) and larger than *Ta ELP+/Ta 54934+* cells (E4, red), which were in the same area (D1). (F) A subset of *Ta SIFGa+* cells in the ventral epithelium >10 μm from the rim expressed the *Ta FFNP* prepropeptide (F1, F2). The yellow channel was displayed with gamma=0.75 to enhance the visibility of weakly stained cells. Color separated images displayed with gamma=1.0 (linear) are shown in S9B Fig. Both *Ta SIFGa+/Ta FFNP-* cells (F3) and *Ta SIFGa+/Ta FFNP+* cells (F4, F5) bear a cilium. The *Ta SIFGa+/Ta FFNP+* cells were larger than the *Ta SIFGa+/Ta FFNP-* cells (F6). * p<0.05. Scale bars 20 μm (A1, B1, F1), 10 μm (C1, D1, E1) and 5 μm in all dissociated cells images.

*Ta sPLA2* (Fig 7A–D). The *Ta LF+/Ta astacin-* cells located near the rim of the dorsal epithelium did not express *Ta ELPE* or the DEC-specific intelectin *Ta 60661* (Fig 6B6). Based on the high expression of *Ta LF* and lack of expression of astacin *Ta 26557*, these cells represent *TH1* metacell C41 (S1 Fig) and metacells *TH1* C188, *TH2* C210, and *HH13* C244 (S3–S5 Figs). The cells that co-express *Ta LF* and astacin *Ta 26557* represent *TH1* metacells C42 (S1 Fig) and C185 and C186 (S3 Fig), *TH2* C215 (S4 Fig), and *HH13* C251 and C252 (S5 Fig). No ortholog

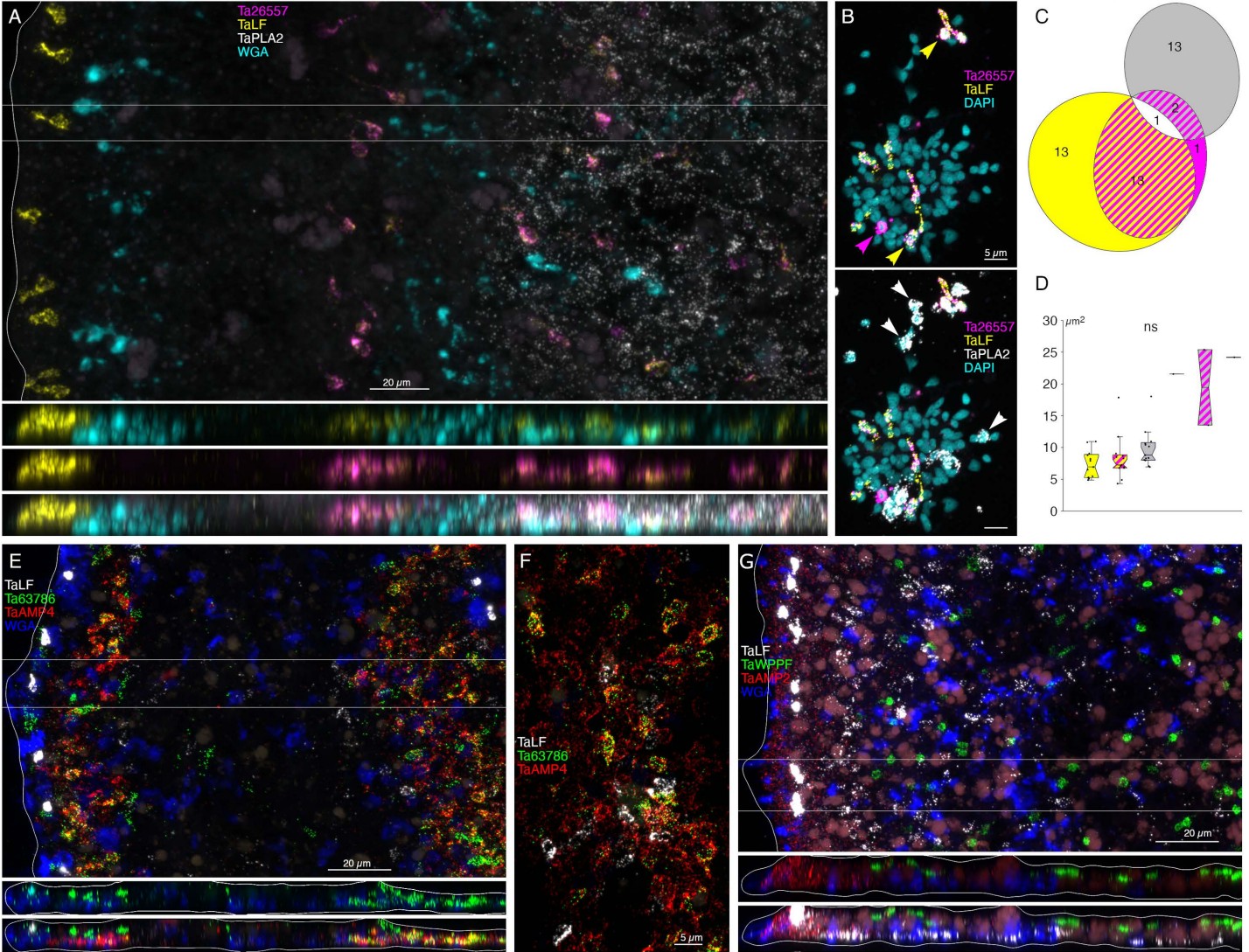

**Fig 7. Localization and characterization of peptidergic cells.** (A) *Trichoplax* wholemount (top panel, horizontal maximum intensity projection; bottom panels, vertical projections of boxed region). Mucocytes were labeled with fluorescent WGA. Cells co-expressing *Ta LF* and astacin *Ta 26557* are interspersed among mucocytes and *Ta PLA2*+ cells in the central part of the ventral epithelium while cells that express *Ta LF* prepropeptide without astacin are most prevalent close to the rim. (B) Dissociated cell preparation. Nuclei were labeled with DAPI. Color separated and merged views show multiple cells that co-express *Ta LF* and astacin *Ta 26557* (yellow arrowheads), a cell that expresses astacin without *Ta LF* (magenta arrowhead) and cells that express *Ta PLA2* (white arrowheads). (C) Many cells co-express *Ta LF* and astacin *Ta 26557*, but few cells co-express *Ta LF+* or *Ta 26557 and Ta PLA2*. (D) All labeled cells are similar in size with *Ta PLA2+*/*Ta 26557+* cells slightly larger. (E) Horizontal and vertical maximum intensity projections of a wholemount labeled with probes for *Ta LF* prepropeptide, secretory protein *Ta 63786*, putative antimicrobial peptide *AMP4* and fluorescent WGA. The images span nearly the entire width of the animal – note the mucocyte and the *Ta LF+* cell at the upper right. Cells in a region of the ventral epithelium within 10 to 40 μm of the rim co-express *Ta 63786* and *AMP4*; cells in the peripheral part of the dorsal epithelium express Ta LF or Ta 63786 but do not express *AMP4*. (F) Enlarged view of partially dispersed cells co-expressing *Ta 63786* and *AMP4*. (G) Horizontal and vertical maximum intensity projections of an animal labeled with probes for *Ta LF*, *Ta WPPF* prepropeptides, *AMP2*, and the lectin WGA. The images encompass ~1/3 the diameter of the animal. Cells expressing *Ta WPPF* prepropeptide are distributed in the dorsal epithelium in a region starting ~40 μm from the rim and do not express *Ta LF* or *AMP2*. Scale bars 20 μm in whole mount images and 5 μm in dissociated cells images.

of LF prepropeptide was found in *CH23* but metacell C233 contained an ortholog of astacin *Ta 26557* (S6 Fig). This metacell contained prepropeptide genes that bore no resemblance to *Ta LF* prepropeptide.

Mucocytes identified by labeling with WGA conjugated to a fluorescent dye were in the ventral epithelium in a zone 10 and 30 μm from the rim and in the central region >50 μm

from the rim (Fig 6C1), as reported [18]. A probe for an oligosaccharide binding protein, *Ta 63702*, labeled both peripheral and central mucocytes (Fig 6C1). A probe for the Arg-Trp-amide (*Ta RWa*) prepropeptide labeled the central mucocytes but not the peripheral mucocytes (Fig 6C1–3). Dissociated mucocytes were ovoid cells ~8 μm in length and lacked a cilium, as evident from the absence of staining for acetylated tubulin (Fig 6C2). Based on co-expression of *Ta 63702* and the *Ta RWa* prepropeptide, the central population of muco-cytes corresponds to *TH1* metacell C36 (S1 Fig), *TH2* metacells C195 and C196 (S4 Fig), *HH13* metacells C213, C216, and C277 (S5 Fig) and *CH23* metacell C209 (S6 Fig). These metacells express PAM, the enzyme that converts a glycine at the C terminal of a peptide to an amide, a secreted glycoprotein with VWD, TIL, C8 and PTS domains characteristic of gel-forming mucins[37], and a secretory protein with no known functional domains, *Ta 62229*, that is not expressed in other metacells. *Trichoplax* H2 metacell C196, *HH13* metacell C218, and *CH23* metacell C210 contain orthologs of *Ta 63702* and the gel-forming mucin but lack the RWa prepropeptide gene, express very little PAM and express only low levels of *Ta 62229*. These metacells likely represent the RWa negative mucocytes that are in the peripheral part of ventral epithelium. Expression of *Ta RWa* prepropeptide was not reported in *TH1* in the more recent scRNAseq study [6]. However, *TH1* metacells C165 and C166 contain ortho-logs of Ta 63702 and the gel-forming mucin (S3 Fig) suggesting that these metacells represent mucocytes. Metacell C165 contains PAM and *Ta 62229* but C166 does not, suggesting C165 represents central mucocytes and C166 peripheral mucocytes. The metacells representing mucocytes in *TH2*, *HH13* and *CH23* contain orthologs of a homeobox DBX1-B-like tran-scription factor that is not present in any other metacells. The GenBank sequence for the *TH1* ortholog of DBX1 is incomplete and consequently the gene was not included in the scRNAseq data for *TH1*.

Cells labeled by probes for the Ta-endomorphin-like peptide (*TaELP*) prepropeptide [22] (S8 Text) and a secreted astacin-like metalloendopeptidase, *Ta 54934*, were in a narrow zone in the ventral epithelium 10–20 μm from the rim (Fig 6D1). The positions of these cells corre-spond with those classified as Type 1 gland cells in ultrastructural studies of *Trichoplax* [18]. Type 1 gland cells contain larger and more electron dense granules than Type 3 gland cells. Dissociated *TaELP* -expressing cells had a cilium (Fig 6D2, D3), nearly always co-expressed both genes (Fig 6D4), and were relatively small (<5 μm in diameter, Fig 6D5). The *TH1* meta-cell that expresses *TaELP* prepropeptide and the astacin *Ta 54934* (C32, Fig 1) highly expresses an additional astacin-like gene, *Ta 54935*, that is not expressed in any other metacells. *TaELP* expression was not reported in *TH1*[6], but a metacell expressing *Ta 54934* and *Ta 54935* was found: C183 (S3 Fig). A metacell expressing *TaELP* ortholog was reported in *TH2*: C210. However, H2 C210 did not express astacin orthologs *Ta 54934* and *Ta 54935*. Instead, these astacins were expressed in H2 C214 (S4 Fig). Moreover, H2 C210 highly expresses LF prepro-peptide, which we found was expressed in distinct sets of cells (Fig 5B1). We suspect that H2 metacell C214 may represent *TaELP* expressing cells. *Hoilungia H13* metacells C248 and C249 expressed orthologs of the *TaELP* prepropeptide and both astacin-like metalloendopeptidases (S5 Fig). No ortholog of *TaELP* prepropeptide was found in *CH23*. However, *CH23* metacell C232 expresses genes identical to astacins *Ta 54934* and *Ta 54935*. Interestingly, metacells that express these astacins in all four species also express one or two G-protein coupled receptors (GPCR) with a C-type lectin domain (CTLD) [35].

A probe for the Ser-Ile-Phe-Gly-amide (*Ta SIFGa*) prepropeptide labeled cells in the dorsal epithelium near the rim and scatted cells in the dorsal and ventral epithelium further in the interior (Fig 6E1). The metacell expressing high levels of *Ta SIFGa* prepropeptide (C46; Fig 1) was reported to express a secreted protein with no known functional domains, *Ta 60437*, that was not expressed in any other metacell. Our probe for *Ta 60437* did not label *Ta SIFGa*-expressing cells but instead labeled a row of cells in the ventral epithelium 20–30 μm

from the rim (Fig 6E1). The positions of these cells correspond with those of Type 1 gland cells. Both cell types possessed a cilium (Fig 6E2, E3). *Ta 60437*+ were larger than cells expressing *Ta SIFGa* (p = 3.32E-4, Bonferroni corrected p value) and cells co-expressing *Ta ELP* and astacin-like metalloendopeptidase *Ta 54934* (p = 0.02, Bonferroni corrected p value), which were in the same area (Fig 6E4). This fact, along with the location of *Ta 60437*+ cells, makes it likely that these are Type 1 gland cells, which are larger than other ciliated cells due to their possession of big secretory granules [18]. The *Ta 60437* gene is highly expressed in meta-cells C167 in *TH1* (S3 Fig), C197 in *TH2* (S4 Fig), C197 and C219 -C222 in *HH13* (S5 Fig) and C111-112 in *CH23* (S6 Fig).

A probe for the Phe-Phe-Asn-Pro-amide (*Ta FFNPa*) labeled the *Ta SIFGa*+ cells in the central part of the ventral epithelium, but not those in the peripheral part of the ventral epithelium or in the dorsal epithelium (Fig 6F1, F2). The positions of the cells that co-express these genes correspond with those classified as ventral Type 3 gland cells [18]. Both *FFNPa+ and FFNPa-* cells possessed a cilium (Fig 6F3-F5), but the *FFNPa*+ cells were larger than *the FFNPa-* cells (p=7.86E-06; Fig5F6). Cells expressing *Ta SIFGa* and *Ta FFNPa* correspond to metacell C38 (S1 Fig) and C168 (S3 Fig). No expression of SIFGa was reported in the metacell expressing FFNPa in *TH2* (S4 Fig), but *HH13* metacell C223 and *CH23* metacell C213 express both SIFGa prepropeptide and very high levels of FFNPa prepropeptide (S5 and S6 Figs). The SIFGa+ cells in the dorsal epithelium and in the rim of the ventral epithelium correspond to *TH1* metacells C46 (S1 Fig) and C189 (S3 Fig), *TH2* C217 (S4 Fig), *HH13* C253-254 (S5 Fig), and *CH23* C221 (S6 Fig) based on high expression of SIFGa prepropeptide and absence of FFNPa prepropeptide.

A probe for the prepropeptide precursor of the peptide WPPF labeled small (diameter ~5 μm) cells distributed throughout the central part of the dorsal epithelium starting ~40 μm from the rim (Fig 7G). Cells expressing *Ta WPPF* represent metacell *TH1* C181 (S3 Fig), *TH2* C213 (S4 Fig), *HH13* C246 (S5 Fig), and *CH23* C230 (S6 Fig).

## Roles of lipophil cells and digestive gland cells in feeding

Both main secretory cell types in the ventral epithelium, LC and ciliated VEC, have been implicated in feeding. Lipophil cells secrete granules whose content lyses algae [4]. VEC constitutively pinocytose extracellular macromolecules such as ferritin [38], HRP, dextran [39] and mucus [18] and, likely, the contents of lysed algae. The expression of high levels of *Ta Trypsin*, *Ta Chymotrypsin* and *Ta sPLA2* in centrally located VEC led us to investigate whether they secrete digestive enzymes during feeding.

We used a fluorescent indicator for trypsin activity, BZiPAR [40], to test whether diges-tive gland cells secrete trypsin during feeding episodes. To visualize LC granule secretion, we supplemented seawater with a lipophilic dye, LipidTOX, which stains intact granules, and the membrane dye FM1–43, which stains the contents of secreted granules [4]. Animals encoun-tering *Rhodomonas salina* algae on the substrate manifested behaviors typical of *Trichoplax* during feeding episodes [4]: they ceased gliding and changing shape, and their margins spread and became more closely attached to the substrate (Fig 8A and S10 Fig; S11 and S12 Movies). Some, although not all, LC in the vicinity of algae released their large apical secretory gran-ule, the contents of which became stained with FM1–43 (Fig 8B, F and S10 Fig). The stained content initially was a diffuse cloud but then became more concentrated. When a LC released its granule, the surrounding area of the ventral epithelium moved farther away from the substrate. These movements, together with the slow scan speeds (~1 frame/sec) used for these experiments, made it difficult to visualize secretion of LipidTOX-stained granules, which happens rapidly, so we used the appearance of FM1–43-stained spots to monitor granule

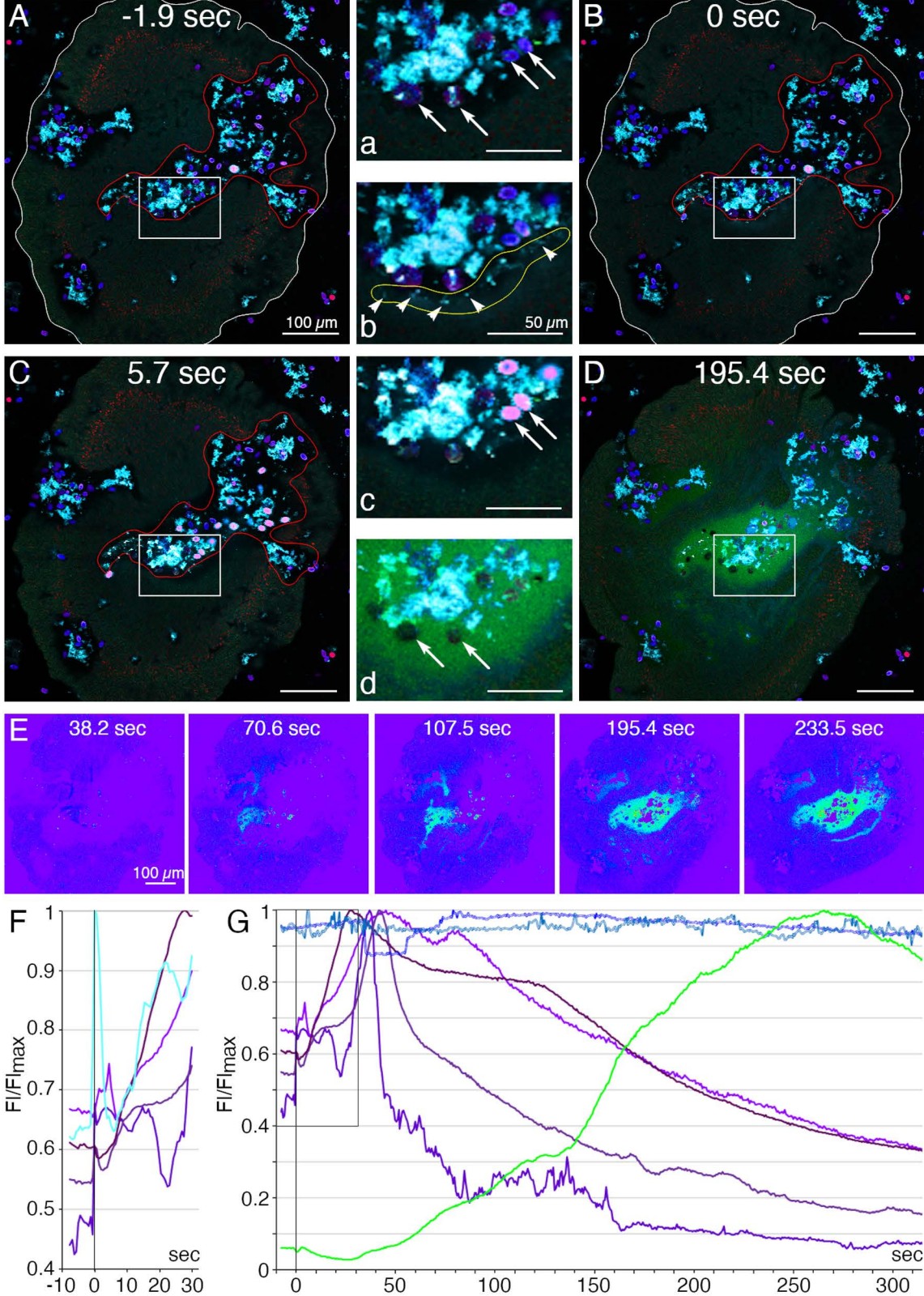

**Fig 8. Secretory behaviors of *Trichoplax* associated with external digestion of algae.** (A–E) Confocal images of a *Trichoplax* feeding on *R. salina* algae; (F, G) normalized fluorescence intensity measured in selected regions. A starved animal was transferred to a cover glass chamber containing algae in seawater with a lipophilic dye (LipidTOX, red) that stains lipophil cell granules; FM1-43

(cyan), a membrane dye used here to visualize the secreted contents of lipophil cell granules; and a fluorescent indicator for trypsin activity (BZiPAR, green). The insets numbered with lowercase letters are enlarged view of rectangular region on a respective image numbered with an uppercase letter. (A, a) At t=-1.9 sec, the animal had ceased moving in a region containing algae (blue, phyco-erythrin autofluorescence, arrows) and debris (cyan) representing algae remnants from an earlier feeding episode. The animal body is outlined white, and the feeding pocket is outlined red. (B, b) At t=0 sec, lipophil cell granule secretion was evident in the feeding pocket due to the sudden appearance of small (<5 μm) diffuse clouds and bright particles (arrowheads on b) of FM1-43-stained material. (C, c) At 5.7 sec, some algae near sites of lipophil granule secretion swelled, lysed, and became intensely stained with LipidTOX and FM1-43 (pink in merged, arrows in c). Other algae (blue) remained intact and were not stained with LipidTOX or FM1-43. (D, d) By 195.4 sec, diffuse BZiPAR fluorescence (green) filled the feeding pocket. The lysed algae no longer were visible, but the intact algae remained (arrows in d). (E) Intensity encoded BZiPAR fluorescence images show increasing trypsin activity between 70.6 and 233.5 sec. (F) Details of first events in feeding: Lipophil granules (cyan; the fluorescence profile obtained for the area outlined yellow in b) were secreted approximately synchronously at 0 sec and this was followed by a rapid increase of fluorescence in nearby algae (four fluorescence profiles obtained for four pink algal cells in c; different shades of magenta). (G) Evidence of digestion: algae affected by lipophil granules (different shades of magenta) burst and released their content. Secretion of trypsin, as indicated by BZi-PAR fluorescence (green; fluorescence profile obtained for the region outlined red in B), began 40–50 sec after lipophil discharge and was associated with a decline of fluorescence intensity of the lysed algal cells. Those algae not affected by lipophil granules remained constant in intensity throughout the feeding episode (two fluorescence profiles obtained for two individual algal cells; different shades of blue). Scale bars 100 μm.

secretion. Some animals contracted during lipid granule secretion and their rims detached from the substrate, but then they flattened and reattached (not illustrated). Thereafter, the rim remained close to the substrate, but the central portion of the animal moved further away, forming a "feeding pocket" between the lower surface and the substrate. The secretion of a granule often was followed within <2 sec by lysis of algae in the vicinity (Fig 8C, F). As algae lysed, they became intensely stained with LipidTOX and FM1–43. Following lysis of algae, cells in the central region of the animal began to display "churning" movements, slow swirling movements of large groups of cells and, sometimes, faster oscillations of small groups of cells (S11 and S12 Movies) [4]. Trypsin activity monitored with BZiPAR became detectable ~ 1 minute after LC granule secretion and continued to increase over the course of 3–10 minutes (Fig 8D, E, G; S10 Fig; S11 and S12 Movies). Algae that had been lysed by the content of LC granules gradually disappeared but algae that were not lysed remained intact (Fig 8G). Lysed algae that became trapped under the rim and therefore were protected from enzymes in the feeding pocket did not disappear, implicating digestive enzymes in the disappearance of the lysed algae. The partially digested content of the lysed algae likely was pinocytosed by VEC and transferred to lysosomes for intracellular digestion [41]. When the animal resumed gliding, the remaining BZiPAR fluorescence diffused from the feeding pocket into the surrounding seawater. Some animals pausing over algae did not secrete LC granules and, consequently, no algae were lysed. Although these animals displayed movements typical of animals feeding on algae, no trypsin activity was detected at their lower surfaces (not illustrated).

## Discussion

We used FISH probes for genes specifically expressed in metacells identified by analysis of single cell transcriptomes of *Trichoplax adhaerens* H1 (*TH1)* [6,30] to map the distributions of different cell types in *TH1* whole mounts and to identify the cells in dissociated cell cultures where their morphology could be visualized by differential interference contrast microscopy and fluorescent labels used to identify hallmark features. We found that two cell types in the central part of the ventral epithelium, lipophil cells (LC) and a subset of the monociliated ventral epithelial cells (VEC), participate in lysis and digestion of microalgae in animals pausing to feed. The dorsal epithelium and the peripheral part of the ventral epithelium contain secretory cells that express genes with sequences resembling precursors of glycoproteins and/or secretory peptides implicated in defense in other animals. Electron microscopic studies of LC, dorsal epithelial cells (DEC) and VEC revealed ultrastructural features of their secretory

granules that, together with the transcriptomes of these cell types, provide clues about the compositions and biogenesis of their secretory granules. We identified metacells representing mucocytes and found that mucocytes in the central part of the ventral epithelium differ from those in the periphery. We mapped the distributions and identified morphological features of seven peptidergic cell types. Our findings provide a more detailed picture of the placozoan body plan (Fig 9) than was apparent from previous microscopic studies and reveal cellular mechanisms placozoans use to obtain nutrients and clues about their defenses against pathogens. We compare the sequences of the main secreted proteins and expression profiles of single cell transcriptomes of four placozoan species and confirm a close relationship between *Trichoplax adhaerens* H1 and H2, partial divergence of *Hoilungia hongkongensis H13* and greater divergence of *Cladtertia collaboinventa H23*.

## A subset of monociliated ventral epithelial cells secrete digestive enzymes

Probes for *Ta trypsin*, *Ta chymotrypsin* and *Ta sPLA2*, which are highly expressed in metacells classified as "digestive gland cells" [6,30], labeled cells in the central part of the ventral epithelium with morphological features typical of ventral epithelial cells (VEC): they possessed an apical cilium and microvilli and had electron dense secretory granules near their apical surface. Although genes associated with cilia were not reported in these metacells, we found that they express genes implicated in cilia structure and function.

We used fluorescent dyes that label the content of lipophil cell granules and a fluorescent indicator for trypsin activity to monitor secretion by lipophil cells (LC) and VEC in *Trichoplax* while they were consuming algae. After the animal crawled onto a patch of algae and ceased moving, LC in the vicinity of algae secreted a large granule whose content lysed nearby algae within less than 2 seconds[4]. Activation of the fluorescent trypsin indicator became detectable after about one minute and the fluorescence increased over the duration of the feeding episode. Detection of trypsin activity provides indirect evidence that central VEC (cVEC) secrete digestive enzymes during feeding episodes. Lysis of algae apparently is required to allow the digestive enzymes to penetrate algae because intact algae were not decomposed even when exposed to digestive enzymes in the feeding pocket.

*Trichoplax* VEC are thought to take up nutrients and further digest them intracellularly because markers such as ferritin and fluorescent dextran were observed inside pinocytotic and endosomal vesicles in VEC after addition to the ambient seawater [38,39]. The gastrodermis in many animals with internal digestive systems contains absorptive cells, called enterocytes, that take up nutrients released from food partially digested by gland cells enzymes[41]. In *Trichoplax*, cVEC appear to combine functions of digestive gland cells and absorptive cells; this could be the reason why little similarity was found at the transcriptomic levels between placozoan digestive gland cells and digestive gland cells in other species [6].

## Lipophil cells – a placozoan synapomorphy?

The observation that the content of granules secreted by LC during feeding episodes in *Trichoplax* rapidly lyses algae led us to search LC metacells for genes that might encode proteins with sequences resembling those of pore-forming proteins [42–45]. We found identical genes in *TH1* and *TH2* LC metacells that encode a protein with a signal peptide and no transmembrane domain; this protein has no orthologs in the NCBI database of non-redundant protein sequences and bears no known domains. The gene is the most highly expressed gene in *TH2* LC metacells, and the second most highly expressed gene in *TH1* LC metacells. The predicted protein is mainly composed of helical regions with positively charged areas that might interact with membranes, like the alpha-helical class of pore-forming proteins [46],

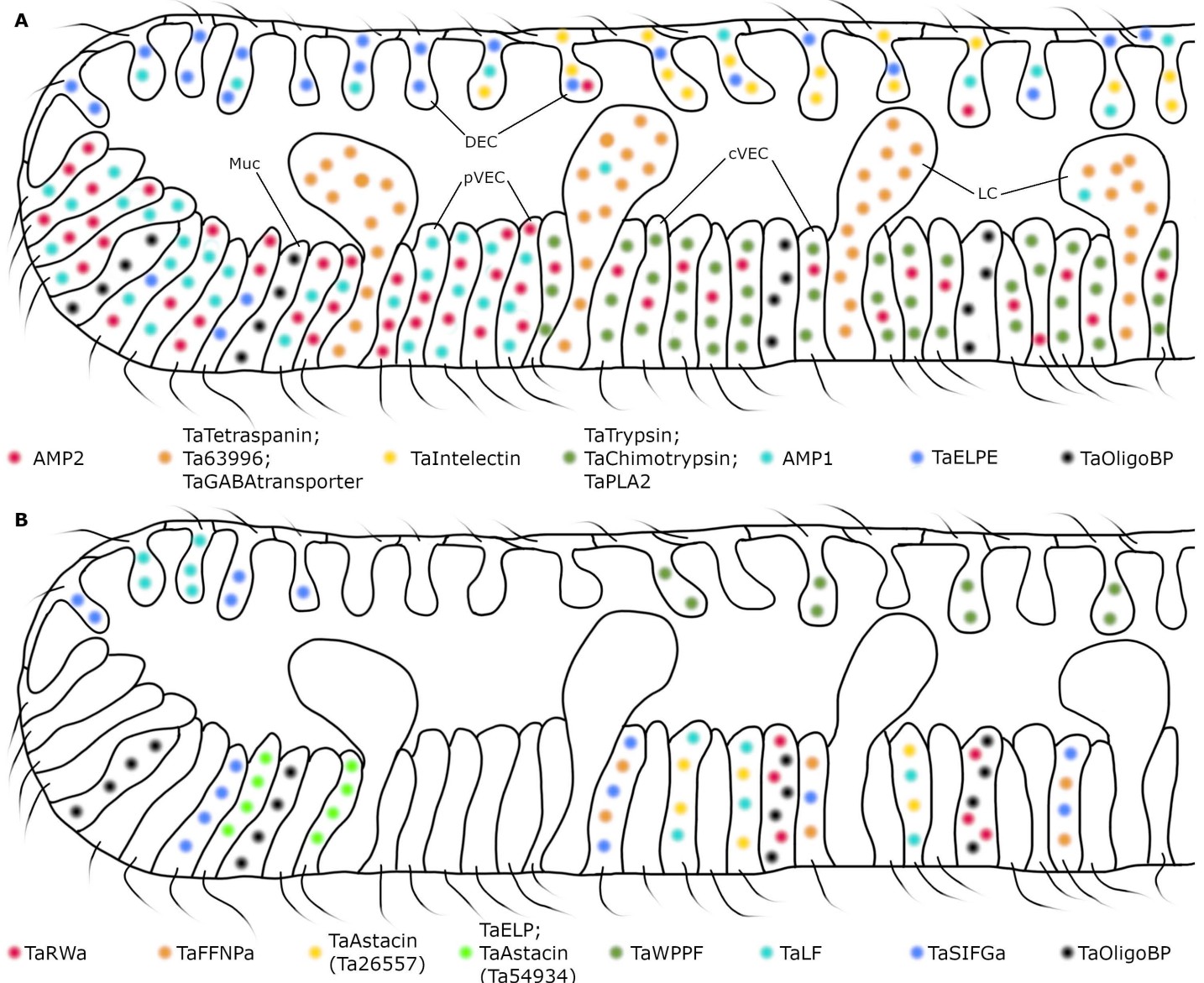

**Fig 9. Summary diagrams of distributions of *Trichoplax adhaerens* secretory cell types.** (A) Secretory proteins, expressed specifically in DEC, pVEC, cVEC, mucocytes, and lipophil cells (peptidergic cells represented in panel B are omitted in panel A). Many central DEC co-express *Ta Intelectin* and the precursor of ELPE peptide but peripheral DEC do not express *Ta Intelectin*. Peripheral VEC highly express precursors of putative antimicrobial peptides (AMP's), while cVEC highly express digestive enzymes. Lipophil cells and mucocytes differentially express their own specific genes (see legend). (B) Secretory products of peptidergic cell types. Localization of peptidergic cell types is based on measurements from animals labeled with *FISH* probes for peptide precursors (Figs 6, 7). Peptidergic cells located in the central zone of the animal co-express additional secretory products unlike the cells located at the periphery. Only mucocytes in the central zone co-express *TaRWa*. Diagrams represent vertical cross sections of a region encompassing ~20% of the diameter of an animal ~500 μm in diameter. Cell dimensions are based on measurements from animals prepared for transmission electron microscopy by high pressure freezing and freeze substitution [16].

nucleoporins [47], or ninjurin [48]. A protein with high sequence similarity to the *TH1*/H2 genes is expressed in one of the two metacell clusters classified as LC in *HH13* (LC2) and an ortholog (e$^{-45}$) is expressed in both LC1 and LC2 metacells in *CH23*. It is likely that this secretory protein is packaged in LC granules and participates in permeabilizing prey organisms. A thin section from a LC revealed an apical granule with an open fusion pore, suggesting that LC granule secretion occurs by conventional exocytosis.

The outer part of the LC apical granule binds osmium, as evident by electron microscopy in thin sections from samples fixed with osmium, indicating the presence of unsaturated lipids [31]. In thin sections of LC from frozen and freeze substituted samples, the apical granules contained variable amounts of electron dense material as well as small vesicles whose content resembled cytoplasm. Granules deeper in the cell bodies of LC contained progressively less electron dense material and larger membrane enclosed vesicles as well as finger-shaped protrusions from the membrane surrounding the granule. The ultrastructure of LC apical granules bears some resemblance to that of lytic granules in rat natural killer cells (NKC), which likewise contain electron dense cores and small membrane-enclosed vesicles [49–51]. Natural killer cell granules are considered secretory lysosomes or lysosome related organelle (LRO). The affinity of acidophilic Lysotracker dyes for LC granules demonstrates that their content is acidic [4,16], like the content of NKC granules.

The presence of lipids in LC granules is intriguing and not a feature shared with NKC granules, although LRO in other types of cells sometimes contain lipids [52,53]. Lipophil metacells highly express several fatty-acid binding proteins [6] that might serve as chaperones to deliver lipids to their granules. *Trichoplax* can lyse and consume cyanobacteria in addition to microalgae [4], indicating that the content of LC granules can penetrate both bacterial and eukaryotic membranes. The identities of the lytic components and their molecular targets in microalgae and cyanobacteria are important questions that remain to be addressed.

The gene expression profiles of LC appear unlike those of cells in the digestive systems of other animals [6]. However, many animal lineages have evolved unique types of secretory cells that they use to capture, incapacitate, or digest prey and to defend against predators. Well documented examples include cnidarian cnidocytes [54–57], ctenophore colloblasts [56,58,59], and venom secreting cells in arthropods, for example [60,61].

## Cells in the peripheral part of the ventral epithelium express putative antimicrobial peptides

Metacells classified as "epithelial" or "lower epithelial" in scRNAseq studies of *TH1* [6,30] based on expression of genes associated with the structure and function of cilia highly express genes with structural similarities to precursors of arminins, antimicrobial peptides found in *Hydra* endodermal cells [32,62,63]. Probes for the mRNA encoding two of these arminin-like genes (referred to here as *AMP1* and *AMP2*) labeled monociliated cells in the peripheral part of ventral epithelium (pVEC). These cells had ultrastructural features like digestive cVEC but possessed darker and more numerous secretory granules. All *TH1* metacells classified as "lower epithelial" express both *AMP1* and *AMP2* and a third arminin-like gene, *Ta 60631 (AMP3)*. Metacells classified as "lower epithelial" in *TH2* express genes that are nearly identical to *AMP1* and *AMP3* and a subset of them express an ortholog of *AMP2*, but *HH13* and *CH23* "lower epithelial" metacells express only an ortholog of *AMP2*.

A probe for a different arminin-like *Trichoplax* gene, *AMP4*, labeled cells in a narrow zone in the ventral epithelium ~15–30 μm from the rim. Their positions correspond to those of cells classified as Type 1 gland cells in ultrastructural studies of *Trichoplax* [18]. The metacells that contain *AMP4* also express genes encoding an intelectin and an uncharacterized secretory protein, *Ta 63786* that were present only in this metacell cluster.

The genes encoding *AMP1*, *AMP2*, and *AMP3* are the three most highly expressed genes in *Trichoplax* H1 "lower epithelial" metacells and the *TH2* orthologs of *AMP1* and *AMP3* are the most highly expressed genes in *TH2* "lower epithelial" metacells. Similarly, arminins are among the most highly expressed genes in most *Hydra* species that have been studied [62,63]. Labeling with mRNA probes for nine arminin paralogs in Hydra wholemounts showed

that eight were expressed in the gastrodermis (endoderm) and one in the epidermis. *Hydra vulgaris* metacells containing arminins were classified as "endodermal epithelial" [64] and likely correspond to absorptive gastrodermal cells (enterocytes). Likewise, AMP expression in *Trichoplax* is mostly located to the ventral epithelium, an anatomical structure involved in digestion and presumably corresponding to cnidarian gastroderm[65].

## Two types of mucocytes arrayed in distinct patterns in the ventral epithelium

We identified metacells representing mucocytes in whole mounts and dissociated cell cultures of *TH1* based on labeling with fluorescent WGA and a probe for an oligosaccharide binding protein, *Ta 63702*, that was highly expressed in a single *TH1* metacell (C36) [30]. Mucocytes were abundant in the peripheral part of the ventral epithelium and more sparsely distributed further in the interior, as reported [18]. Mucocytes in the central region of the ventral epithelium, but not those in the periphery, co-expressed a gene for prepropeptide that is predicted to produce RWamide peptides. We identified metacells in *TH2*, *HH13* and *CH23* that express a gene identical to *Ta 63702* and a subset of these metacells contain a gene encoding the RWamide prepropeptide. Although no *Ta RWa* gene was found in the second RNAseq analysis of *TH1* [6], we identified metacells that likely represent central and peripheral mucocytes in this dataset. All metacells representing mucocytes in the more recent RNAseq datasets contain a gene encoding an intelectin (Ta 62229) and a protein with VWD, TIL, C8 and PTS domains characteristic of gel-forming mucins [37]. The RWamide peptides and intelectin likely are packaged and secreted along with mucus since mucocytes contain only one type of secretory granule [18]. Animal mucosa often contain lectins and antimicrobial peptides [66]. However, the predicted products of the *Ta RWa* do not have electrostatic properties typical of antimicrobial peptides. It is possible that the peptides are used for intercellular signaling, although synthetic RWa peptides had no apparent effect on the behavior of *Trichoplax* when added to their culture dishes [23]

Genes encoding gel-forming mucins with VWD, TIL, C8 and PTS domains were found in single cell transcriptomes of the cnidarian *Hydra vectensis* and the ctenophore *M. leidyi* [30,67]. Mucus-like substances have been observed on epithelia of adult demosponges [68] and cells resembling mucocytes were found by electron microscopy in metamorphosing larvae of several classes of sponges[69–71]. The distribution of mucus-secreting cells across animal phyla and presence of mucus-like genes in Choanoflagellates [72] suggests that the common ancestor of Metazoa may have possessed cells that secreted a mucus-like substance.

## Dorsal epithelial cells secrete lectins implicated in defense

Cells with morphological characteristics of DEC were labeled with a probe for mRNA encoding the ELPE prepropeptide and many of them were labeled with a probe for an intelectin, *Ta 60661*. The *TH1* metacells expressing these genes were classified as "epithelial" or "upper epithelial" [6,30] and include genes encoding ten intelectins that are not expressed in any other metacells. *Trichoplax* H2 "upper epithelial" metacells contain identical intelectin genes. *Hoilungia* H13 "upper epithelial" metacells contain nearly identical orthologs of five of them and *CH23* "upper epithelial" metacells contain four of them. These intelectins likely are packed inside the electron dense granules in DEC, like intelectins in goblet cells in mammalian intestine [73]. These metacells also express a gene encoding a membrane associated mucin 4-like glycoprotein that may be a component of the glycocalyx observed on the apical surfaces of DEC by electron microscopy.

Intelectins in chordate epithelia are thought to protect against invasion by pathogens by binding glycans on their surfaces and aggregating the cells [36]. Less is known about the prevalence and functions of intelectins in non-chordates, although the available evidence suggests that the primary function is defense [36,74].

## Peptidergic cell types

We mapped the positions and studied the morphology of cells expressing seven of the fourteen prepropeptides identified by scRNAseq in *TH1* [6]. Peptides that are the predicted/possible products of the six of these prepropeptides (WPPF, ELPE, SIFGa, FFNPa, LF, TaELP) elicited changes in the behavior of *Trichoplax* when added to their culture dishes[22,23]. We anticipated that the positions of the peptidergic cells and the responses elicited by peptides they secrete might provide clues about their functions.

Cells that expressed the *Ta SIFGa* prepropeptide were most prevalent in a narrow zone near the rim of the dorsal and ventral epithelium but were also present in more central regions of the epithelia. The *Ta SIFG+* cells in the central part of the ventral epithelium co-expressed the *Ta FFNPa* peptide precursor. Addition of low concentrations (<500 nM) of SIFGa to dishes containing *Trichoplax* rapidly (<1 min) elicited contraction of the entire animal. Higher concentrations of the peptide caused the animal to fold up and detach from the substrate[22,23]. The peptide FFNPamide elicited an expansion of the area of the animal and an increase in the frequency of spontaneous pauses in movement. The latencies of the responses to FFNPa were longer (> 7 minutes) than those of the contractions elicited by SIFGamide, suggesting that simultaneous secretion of SIFGa and FFNPa by cells that co-express these peptides might elicit contraction followed by relaxation. Detachment from the substrate elicited by secretion of endogenous SIFGa may allow the animal to move between the substrate and air/water interface, as animals maintained in culture often do. The location of the cells that co-expressed *Ta SIFGa* and *Ta FFNPa* suggests that they may participate in orchestrating behaviors that occur during feeding.

Cells expressing the gene for LF prepropeptide were in a narrow zone near the rim of the dorsal epithelium and distributed throughout the central part of the ventral epithelium. The *Ta LF+* cells in the ventral epithelium co-expressed a secreted astacin-like metalloendopeptidase. Applying LF to *Trichoplax* elicited a large expansion in the animals' area. In addition, animals that were translocating stopped and rotated in place [23], indicating a change in the collective behavior of the monociliated VEC that propel crawling. Secretion of endogenous LF by cells located at the rim could promote adhesion of the rim to the substrate by relaxing the contractile cytoskeletons of cells in the rim. The location of the cells that co-expressed *Ta LF* and astacin *Ta 26557* suggests that they may have roles in feeding.

Cells co-expressing the *TaELP* prepropeptide and an astacin-like metalloendopeptidase were in the ventral epithelium near the rim. Their locations did not correspond to those of cells labeled with antiserum against *TaELP* prepropeptide or against the peptide YPFFamide (human endomorphin 2), which instead labeled mucocytes[18,22]. We previously reported that adding YPFFamide or QDYPFFamide to a dish containing moving *Trichoplax* consistently caused them to stop moving and arrested ciliary beating, but the predicted products of the *TaELP* prepropeptide (QDYPFFGN or pQDYPFFGN) arrested movement of fewer than half of the animals tested [22]. The observation that antiserum against YPFFamide did not label the cells that express *TaELP* prepropeptide suggests that these cells do not produce the peptide QDYPFFamide and casts doubt on the idea that the function of these cells is to detect food and arrest movement during feeding episodes[22]

The *Ta ELPE* prepropeptide was expressed in a large fraction of DEC and by cells near the rim of the ventral epithelium. Cells expressing the *Ta WPPF* prepropeptide were interspersed among DEC in the central part of the dorsal epithelium. Bath application of ELPE or WPPF were reported to elicit behaviors reminiscent of behaviors observed in animals feeding on biofilms: periodic cessation of gliding accompanied by churning movements of cells in central regions of the ventral epithelium[23]. The ELPE peptide elicited a small expansion in the animals' area, comparable to that observed in animals pausing to feed, whereas the WPPF peptide elicited a much larger expansion. The positions of the cells that express *Ta ELPE* or *Ta WPPF* make it unlikely that their primary function is to initiate feeding behaviors.

We identified orthologs for four of the seven studied prepropeptides across all four placozoan species. Additionally, the TaELPE precursor was detected only in TH1 and TH2. This pattern of phylogenetic distribution and sequence divergence reflects what is observed across cnidarian classes and bilaterian phyla [33,34]. These findings further emphasize the biological divergence among placozoan species, despite their morphological similarities. The morphology and expression profiles of placozoan peptidergic cells closely resemble those of sensory-secretory cells found in Cnidaria and Bilateria [6,16,18]. Further studies on peptide receptors, along with their associated signaling pathways and cellular interactions, are necessary to better understand their biological roles and functions.

## Conclusions

Placozoa provide an opportunity to explore the biology of animals with body plans and life-styles reminiscent of ancestors of present day metazoa [75–80]. The availability of single cell transcriptomes of four placozoan species [6,30] provides a rich resource for investigating the functions and evolutionary histories of their cell types. Building on this foundation, we linked single-cell transcriptomic data with previously described morphological cell types and explore the roles of their secretory products.

Placozoan secretory cell types share morphological features with those found in other metazoans, with several expressing homologs of functionally characterized proteins. The ventral epithelial cells (VECs) in the central part of the animal secrete digestive enzymes, similar to zymogen gland cells in cnidarians and bilaterians. In contrast, peripheral VECs express putative antimicrobial peptides (AMPs), akin to gastrodermal cells in various animals. Both central and peripheral VECs possess a cilium and microvilli and engage in pinocytosis to absorb nutrients released from partially digested food, resembling enterocytes in cnidarians and bilaterians. Defensive epithelial cells (DECs) in placozoans express glycoproteins associated with pathogen defense across metazoans, while mucocytes produce orthologs of gel-forming mucins, a key component of mucus. Additionally, placozoans exhibit a variety of peptidergic cell types, each characterized by distinct distributions and gene expression profiles. Lipophil cells secrete lytic substances like certain secretory cells in other animals. However, the composition of their secretory granules appears unique to placozoans. Collectively, our work highlights the diversity of secretory cells and their products in Placozoa, a phylum with a unique phylogenetic position within the metazoan tree of life. These findings provide valuable insights for comparative studies across animals, advancing our understanding of the evolution of secretory cells.

## Methods

### Animals

*Trichoplax adhaerens* (Schultze, 1883) of the Grell (1971) strain, a gift from Leo Buss (Yale University), were kept in Petri dishes with artificial seawater (ASW; Instant Ocean,

Blacksburg, VA, USA) supplemented with 1% Micro Algae Grow (Florida Aqua Farms, Dade City, FL, USA) and red algae (*Rhodomonas salina*, Provasoli-Guillard National Center for Culture of Marine Plankton, East Boothbay, ME, USA), as described previously [16]. Water was partially changed once a week.

## Cell dissociation

To prepare dissociated cells, a group of animals was rinsed in calcium and magnesium free ASW (calcium-free ASW) [81] and then incubated in 0.25% trypsin in calcium-free ASW for 2 h. The animals were transferred to normal ASW and triturated with a glass Pasteur pipette until the suspension was homogeneous.

## Fluorescence *in situ* hybridization

*In situ* hybridization was performed with probes and reagents from Advanced Cell Diagnostics, Inc. (ACD, Hayward, CA, USA) using protocols developed to optimize staining in *Trichoplax* [18,24]. The RNA sequences were retrieved from PubMed or from a *T. adhaerens* transcriptome database, access to which we were granted by Adriano Senatore (University of Toronto). RNAscope probes for multiplex fluorescence *in situ* hybridization were designed by ACD. Catalog names and numbers, gene names, and annotations, and mRNA accession numbers are listed in Table 1. Further details are available in the ACD product catalog.

Animals and dissociated cells were transferred to silanized cover slips or Superfrost Plus Gold glass slides (Thermo Fisher Scientific, Pittsburgh, PA, USA) with a drop of ASW mixed 1:1 with 0.97 M mannitol (in water). Samples were left on cover slips/slides to adhere for about 2 h. Then the liquid was blotted, and the cover slips/slides were plunged into prechilled tetrahydrofuran on dry ice and kept overnight. Wholemount samples were transferred to 3% acetic acid in methanol at -20 °C for 30 min followed by a mixture of formalin and methanol (1:10), initially at -20 °C and then at room temperature (RT) for 30 min. The wholemount samples were rinsed twice in methanol, dried for 5 min and then treated with Protease IV for 30 min. The dissociated cell samples were transferred directly to a mixture of formalin and methanol (1:10), initially at -20 °C and then at RT for 30 min. The cell samples were then rinsed once with methanol, twice with ethanol followed by descending concentrations of ethanol in PBS (70% and 50%) and PBS. The cell samples were treated with Protease III diluted 1:15 in PBS for 15 min. Hybridization was performed with RNAscope Fluorescent Multiplex Reagent Kit (# 320850) according to supplier's instructions. The negative control 3-Plex Negative Control Probe gave no labelling as did the sense probes for two of the genes studied (TaYPFFamide, Cat. #488721; and Ta 26557, Cat #1222671-C3). Samples were counterstained with 1:200 wheat germ agglutinin (WGA) conjugated to Alexa 647 (# W32466, Thermo Fisher Scientific, Waltham, MA) or CF405M (# 29028, Biotium, Hayward, CA, USA) and/or DAPI, and mounted in ProLong™ Gold antifade reagent (# P36934, Invitrogen, Eugene, OR, USA). Dissociated cell preparations were also subjected to immunolabelling for tubulin after RNAscope steps to visualize cilia. For that, samples rinsed with wash buffer after the last (AMP4) hybridization step were further washed twice with PBS (pH7.4) and once with blocking buffer (BB, 3% goat serum, 2% horse serum, 1% BSA in PBS). Then, primary anti-acetylated tubulin mouse antibody (#T7451; Sigma) diluted 1:500 in BB was applied. After overnight incubation with the primary antibody, samples were washed with PBS and incubated in secondary Atto-655 goat anti-mouse antibody (#50283; Sigma) or Alexa 555 goat anti-mouse (#A21422; ThermoFisher) diluted 1:200 in blocking buffer. Finally, samples were rinsed from secondary antibody with PBS, briefly incubated in DAPI, and mounted in ProLong™ Gold antifade reagent (ThermoFisher). The samples were examined with a Plan-Apochromat 40X NA 1.3 or 63X NA 1.4 Plan-Apochromat objective on a LSM 800 or

**Table 1. RNAscope probes for multiplex fluorescence *in situ* hybridization.**

| RNAscope probe | Catalog # | Gene | Annotation | Accession number |
|---|---|---|---|---|
| Ta-57870-C2 | 561141-C2 | Ta-57870 | PLA2 | XM_002113901.1 |
| Ta-63128 | 561031 | Ta-63128 | Trypsin | XM_002115568.1 |
| Ta-Chymotrypsin-C2 | 572831-C2 | Ta-63088 | Chymotrypsin | XM_002109248.1 |
| Ta-PhospholipaseA2 | 572841 | Ta-63140 | PLA2 | XM_002116279.1 |
| Ta-57870-C2 | 561141-C2 | Ta-57870 | PLA2 | XM_002113901.1 |
| Ta-56030-C2 | 56030-C2 | Ta-56030 | AMP2 | XM_002111672.1 |
| Ta-55945-C3 | 55945-C3 | Ta-55945 | AMP1 | XM_002111637.1 |
| Ta-64402-C2 | 1313051-C2 | Ta- 64402 | AMP3 | XM_002118041.1 |
| Ta-58643-C1 | 1138211-C1 | Ta-58643 | Tetraspanin | XM_002114763.1 |
| Ta-63786 | 1209181-C1 | Ta-63786 | Ta Sec Prot | XM_002110591.1 |
| Ta-64037-C2 | 1222681-C2 | Ta-64037 | Ta Sec Prot | XM_002113441.1 |
| Ta-63996-C2 | 1138221-C2 | Ta-63996 | Ta Sec Prot | XM_002113590.1 |
| Ta-29105-C3 | 1138231-C3 | Ta-29105 | Tansporter | XM_002115362.1 |
| Ta-63702-C3 | 823661-C3 | Ta-63702 | Oligosac BP | XM_002110164.1 |
| Ta-60185-C2 | 487491-C2 | Ta-60185 | SIFGamide Pre | XM_002116138.1 |
| None | 856541-C2 | EST | RWamide Pre | GR951923.1 |
| Ta-63942-C2 | 823731-C2 | Ta-63942 | FFNPamide pre | XM_002112788.1 |
| Ta-51275 | 561121 | Ta-51275 | LF Pre | XM_002108379.1 |
| Ta-51275-C3 | 561121-C3 | Ta-51275 | LF Pre | XM_002108379.1 |
| Ta-YPFFamide-O1-C2 | 1228931-C2 | AQX36197.1 | TaELP Pre | KY675296.1 |
| Ta-56359 | 561051 | Ta- 56359 | WPPF Pre | XM_002112355.1 |
| Ta-60185 | 487491 | Ta-60185 | SIFGa Pre | XM_002116138.1 |
| Ta-64280-C3 | 8236681-C3 | Ta-64280 | ELPE Pre | XM_002116364.1 |
| Ta-26557-C3 | 561111-C3 | Ta-26557 | Astacin | Tricho_evg1529192 |
| Ta-54934 | 560991 | Ta-54934 | Astacin | XM_002110640.1 |
| Ta-60661-C3 | 1241961-C3 | Ta-60661 | Intelectin | XM_002116619.1 |
| Ta-60661-C2 | 1217761-C2 | Ta-60661 | Intelectin | XM_002116619.1 |
| 3-Plex Neg. control | 320871 | | | |

Abbreviations: Ta – Triaddraft-; PLA2 – PhopholipaseA2; AMP– Putative antimicrobial peptide; Ta Sec Prot – Predicted *T. adhaerens* secretory protein; Oligosac BP – Oligosaccharide binding protein; Pre – Prepropeptide

LSM 880 AIry laser scanning confocal microscope with DIC and fluorescence optics (Carl Zeiss Microscopy, LLC). The areas of dissociated cells visualized with DIC optics were estimated by measuring the length of the long axis of the cell body and the perpendicular axis and calculated using the equation for an ellipse. Cells were considered labeled if they possessed at least two fluorescent grains. To demonstrate expression patterns in wholemounts, we created xy maximum intensity projections in Zen software (Carl Zeiss Microscopy, LLC). In order to show the distribution of a signal across the animal thickness, we made xz projections for selected strip regions, indicated on the xy images with a narrow box. Labelling with each probe was done at least twice; the results of independently repeated experiments were similar.

## Time-lapse microscopy of living animals

To monitor secretion of lipophil cell granules and trypsin during feeding episodes, animals first were transferred to a 35 mm petri dish containing ASW and kept for one to three hours and then incubated for 10 minutes in ASW containing FM1–43 (N-(3-Triethylammoniumpropyl)-4-(4-(Dibutylamino) Styryl) Pyridinium Dibromide; Thermo Fisher;

#T3163; 1μg/ml) and LipidTOX (HCS LipidTOX Deep Red neutral lipid stain; Thermo Fisher; #34477; 3 μl/ml). The animals were transferred to a RC-40LP chamber containing ASW with FM1–43, LipidTOX, BZiPAR (Rhodamine 110, bis-(N-CBZ-L-isoleucyl-L-prolyl-L-arginine amide), dihydrochloride; Biotium; #10208; 8 μM) and *Rhodomonas salina* algae. LipidTOX stains intact lipophil cell granules and FM1–43 stains the contents of secreted granules, as described previously [4]. BZiPAR is a fluorescence indicator for trypsin activity [40] The behavior of the animals was monitored with a 32-channel spectral detector and a 10X NA 0.45 Plan-Apochromat objective on a LSM880 confocal microscope with 488 nm, 561 nm and 647 nm illumination. Images (512 X 512 bits) were captured at ~1.7 frames/second. Reference spectra used for linear unmixing were collected from the following samples: (1) FM1–43, ventral epithelium of an animal labeled with FM1–43; (2) LipidTOX, lipophil cell granule in an animal labeled with LipidTOX; (3) algae; autofluorescence; (4) BZiPAR; seawater containing BZiPAR and 0.25% trypsin. Bicubic interpolation was used to enlarge the images and reduce pixilation. Still images used for illustration in Fig 8 were created by averaging two successive frames to improve signal. Fiji software was used to build fluorescence profiles shown in Fig 8.

## Electron microscopy

For transmission electron microscopy (TEM), scanning electron microscopy (SEM), and electron microscope tomography animals were high pressure frozen, freeze-substituted and embedded as described previously [16]. WGA-nanogold (EY Laboratories, San Mateo, CA, USA) labelled thin sections were prepared as described[18]

For transmission electron microscopy, the ultrathin sections were observed in a JEOL 200-CX (JEOL, Japan) at 120–200 kV and were imaged with an AMT camera mounted below the microscope column. Image processing and color intensity measurement was done with Fiji.

The protocol for serial sectioning followed by SEM imaging in backscatter mode was published elsewhere [18]. Series of these images were used to study the nature of lipophil granules protrusions and to count granules in the pVEC and cVEC.

For EM tomography, we used ~100 nm sections and a transmission electron microscope, JEOL 1400 (JEOL, Japan) at an accelerating voltage of 120kV. Images were taken at ~1° tilt intervals from -69° to 69° at a single tilt axis at the magnification x5000. The tilt images were aligned using an electron tomography software package, EM3D (em3d.org), upgraded by Jae Hoon Jung and Eun Jin Choi in Dr. Reese's laboratory, and reconstructed to generate a reconstructed volume or tomogram by Simultaneous Iterative Reconstruction Technique (SIRT) with 10 iterations using the algorithm implemented in a tomography program, called VEM (Volume Electron Microscopy), developed by Jae Hoon Jung and Eun Jin Choi. The examination of the virtual slices was carried out using both EM3D and VEM. The virtual slices through the tomogram were one-voxel thick, which is 1.8 nm.

To prepare freeze fractured samples, *T. adhaerens* were placed on the surface of a gelatin cushion and directly frozen by contact with a sapphire disk glued to the surface of a copper block cooled to −186 °C using a LifeCell CF-100 slam-freezing machine. Samples were transferred to a Balzer freeze-fracture apparatus. Replicas were prepared by freeze-fracturing the specimen at −110 °C and at a vacuum of $10^{-7}$ Torr, rotary shadowing with platinum and carbon, and backing the replica with carbon. The replicas were cleaned with sodium hypochlorite, mounted on grids, and examined using a JEOL 200-CX at 120 kV.

## Protein/peptide predictions

Deep TMHMM [82] was used to predict signal peptides and transmembrane alpha helices. InterPro online tool [83]was used to predict protein topology and conserved domains.

AlphaFold protein database [84] was used to predict secondary structures of the *Trichoplax* proteins. ChimeraX-1.5 [85,86]was used to render a secondary structure of the proteins and color protein surface by electrostatic potential.

Regions of prepropeptides were annotated and classified into signal peptide region using SignalP6.0 [https://pubmed.ncbi.nlm.nih.gov/34980915/], prohormone convertase cleavage sites (dibasic residues), mature peptide (based on characteristics of cnidarian neuropeptide processing [87]. C-terminal glycine residues were predicted to be processed into amide-groups, and N-terminal glutamates were predicted to be processed into pyroglutamates.

## Quantitative image processing and Statistics

We used Paleontological Statistics (PaSt), freely available software [88], to do statistical comparisons and build boxplots. Mann-Whitney test was applied, since many variables did not follow the Gaussian distribution. For multiple samples, we used an ANOVA test followed by Mann-Whitney post hoc pairwise test with the Bonferroni correction. In boxplots, the box represents 25–75 quartiles, and the *median* is shown with a horizontal line inside the box. "Whiskers" show the minimal and maximal values excluding outliers. Each value is plotted as a dot on top of a boxplot. Accurate area-proportional Venn diagrams were drawn using eulerAPE program [89] or PaSt.

## Gene expression profiles

The data for S1 Fig was downloaded from [30]. The fold change and percentage expression were extracted for the selected genes across all the metacells and visualized using the tidyverse package in R. Gene expression heatmaps were generated from [https://sebelab.crg.eu/placo-zoa_cell_atlas/] using orthologous genes from TH1, TH2, HH13, and CH23. The genes and meta cells were manually annotated using InkScape.

## Supporting information

**S1 Fig. Expression of selected genes across T. adhaerens metacells (columns) identified by single cell RNA sequencing.** Data are from [30]. Cell types are identified based on data from the present study and [6,18,24,30]. Dot color represents fold change of gene expression and dot size represents percentage of the total UMI in the given metacell.
(TIF)

**S2 Fig. Color separated FISH images of the micrographs shown in Fig 2**. (A, B) Separated channels corresponding to horizontal (xy) and vertical (xz) projections of color-merged *FISH* images of TH1 wholemounts in Fig 2A and B. (C, D) Separated DIC and fluorescence channels of the merged images of dissociated cells in Fig 2C and D.
(TIF)

**S3 Fig. Normalized expression of selected genes across metacells for *TH1*.** Cell types are identified based on data from the present study and [6,18,30]. Expression data from: https://sebelab.crg.eu/placozoa_cell_atlas/.
(TIF)

**S4 Fig. Normalized expression of selected genes across metacells for TH2.** Expression data from: https://sebelab.crg.eu/placozoa_cell_atlas/.
(TIF)

**S5 Fig. Normalized expression of selected genes across metacells for HH13.** Expression data from: https://sebelab.crg.eu/placozoa_cell_atlas/.
(TIF)

**S6 Fig. Normalized expression of selected genes across metacells for CH23.** Expression data from: https://sebelab.crg.eu/placozoa_cell_atlas/.
(TIF)

**S7 Fig. Secondary structure predictions and electrostatic potential maps for putative Trichoplax AMPs (AMP1 (A), AMP2 (B), and AMP3 (C) and *Hydra vulgaris* arminin (D).** Secondary structures are predicted with Alpha Fold and peptide surfaces are colored by electrostatic potential with ChimeraX.
(TIF)

**S8 Text. Placozoan prepropeptides, predicted cleavage sites and processing motifs.** Underlined regions represent predicted signal peptides. Highlighted yellow represents the mature peptide region, green represents prohormone convertase cleavage sites, cyan represents pyroglutamate, and magenta represents C-terminal amide.
(DOCX)

**S9 Fig. Color separated FISH images of TH1 wholemounts corresponding to horizontal (xy) and vertical (xz) of color-merged FISH images in Fig 5F (A) and Fig 6F1 (B).**
(TIF)

**S10 Fig. Secretory behaviors of *Trichoplax feeding* on *R. salina* algae.** Lipophil cell granules were labeled with LipidTOX (red). The fluorescent membrane dye FM1–43 (cyan) was added to the seawater to label cell membranes and the contents of LC granules. BZiPAR, a fluorescent indicator of trypsin activity (green), was added to detect secretion of trypsin. Algae are visible by autofluorescence and FM1–43 staining (pink in merged images). At the beginning of the sequence, the peripheral part of the animal was closely attached to the substrate, while the central part was invaginated, forming a feeding pocket enclosing algae. At t=0 sec, lipophil granule secretion was evident due to the appearance of small FM1–43-stained spots (cyan/white; arrowheads) in the feeding pocket. At t=12 sec, several algae (pink) in the feeding pocket were lysed and released material stained by FM1–43 (cyan). By 150–287 sec, trypsin activity was evident in the feeding pocket and the lysed algae were decomposing. Bottom panels show intensity-coded images of BZiPAR trypsin activity indicator at sequential timepoints for the boxed region.
(TIF)

**S11 Movie. *Trichoplax* feeding on *R. salina* algae (same animal as illustrated in Fig 8).** Lipophil cell granules were labeled with LipidTOX (red). The fluorescent membrane dye FM1–43 (cyan) was added to the seawater to label cell membranes and the contents of LC granules. BZiPAR, a fluorescent indicator of trypsin activity, was added to detect secretion of trypsin. Algae are visible by autofluorescence (magenta). Time stamp is at upper left. Lipophil cell granule secretion begins at ~50 sec as evident from the appearance of small FM1–43-stained puncta in the vicinity of algae. Green fluorescence generated by proteolysis of BZiPAR becomes visible at ~110 sec and reaches a peak at ~270 sec. Content of lysed algae is stained by FM1–43, but staining declines as the algae decompose. The rim of the animal remains closely attached to the substrate forming a feeding pocket that confines the contents of the lysed algae and the BZiPAR fluorescence.
(MP4)

**S12 Movie. *Trichoplax* feeding on *R. salina* algae (same animal as illustrated in S10 Fig).** Staining is the same as in S11 Movie. Lipophil granule secretion begins at ~14 sec as evident from the appearance of small clouds of FM1–43-stained material in the vicinity of algae. The clouds coalesce into puncta within seconds. Algae begin to lyse at ~20 sec. Trypsin activity becomes visible at ~45 sec.
(MP4)

## Author contributions

**Conceptualization:** Tatiana D. Mayorova, Thomas Lund Koch, Bechara Kachar, Jae Hoon Jung, Carolyn L. Smith.

**Data curation:** Tatiana D. Mayorova, Thomas Lund Koch, Bechara Kachar, Jae Hoon Jung, Carolyn L. Smith.

**Formal analysis:** Tatiana D. Mayorova, Thomas Lund Koch, Bechara Kachar, Jae Hoon Jung, Carolyn L. Smith.

**Funding acquisition:** Bechara Kachar, Thomas S. Reese, Carolyn L. Smith.

**Investigation:** Tatiana D. Mayorova, Bechara Kachar, Jae Hoon Jung, Carolyn L. Smith.

**Methodology:** Tatiana D. Mayorova, Thomas Lund Koch, Bechara Kachar, Jae Hoon Jung, Carolyn L. Smith.

**Project administration:** Tatiana D. Mayorova, Carolyn L. Smith.

**Resources:** Bechara Kachar, Thomas S. Reese, Carolyn L. Smith.

**Software:** Jae Hoon Jung.

**Supervision:** Tatiana D. Mayorova, Thomas S. Reese, Carolyn L. Smith.

**Validation:** Tatiana D. Mayorova, Thomas Lund Koch, Bechara Kachar, Carolyn L. Smith.

**Visualization:** Tatiana D. Mayorova, Thomas Lund Koch, Bechara Kachar, Carolyn L. Smith.

**Writing – original draft:** Tatiana D. Mayorova, Carolyn L. Smith.

**Writing – review & editing:** Tatiana D. Mayorova, Thomas Lund Koch, Bechara Kachar, Jae Hoon Jung, Carolyn L. Smith.

## Acknowledgements

We thank the Electron Microscopy Facility of the National Institute of Neurological Disorders and Stroke, the Central Microscopy Facility of the Marine Biological Laboratory, Christine A. Winters (NINDS) and Katherine Hammar (MBL) for sample preparations and technical support. We thank Professor Adriano Senatore, University of Toronto, for valuable comments on the manuscript. This work was supported by the Intramural Research Program of the National Institute of Neurological Disorders and Stroke, National Institutes of Health, National Institutes of Health, Bethesda, MD, USA.

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
