## [Decision Letter · Decision Letter 0]

20 Oct 2024

PONE-D-24-38008Placozoan secretory cell types implicated in feeding, innate immunity and regulation of behaviorPLOS ONE

Dear Dr. Smith,

Thank you for submitting your manuscript to PLOS ONE. After careful consideration, we feel that it has merit but does not fully meet PLOS ONE’s publication criteria as it currently stands. Therefore, we invite you to submit a revised version of the manuscript that comprehensively addresses the points raised during the review process.

We look forward to receiving your revised manuscript.

Kind regards,

Michael Schubert

Academic Editor

PLOS ONE

Journal Requirements:

Reviewers' comments:

Reviewer's Responses to Questions

**Comments to the Author**

1. Is the manuscript technically sound, and do the data support the conclusions?

Reviewer #1: Partly

Reviewer #2: Partly

2. Has the statistical analysis been performed appropriately and rigorously? 

Reviewer #1: I Don't Know

Reviewer #2: Yes

3. Have the authors made all data underlying the findings in their manuscript fully available?

Reviewer #1: Yes

Reviewer #2: Yes

4. Is the manuscript presented in an intelligible fashion and written in standard English?

Reviewer #1: Yes

Reviewer #2: Yes

5. Review Comments to the Author

Reviewer #1: This paper by Mayorova and colleagues is an ambitious first attempt at correlating single cell transcriptomics data and morphological data (obtained by electron microscopy and confocal microscopy either by this or other teams) across four species belonging to phylum Placozoa. To achieve this, the authors mostly exploit a multiplex fluorescent mRNA In Situ Hybridization approach on Trichoplax adhaerens H1, and use this as a template for establishing comparison with the single cell transcriptomes from Trichoplax H2 and two Hoilungia haplotypes , H13 and H23.

Such an approach would indeed lead to a very-fine level characterization of the different cell types constituting the epithelia of Trichoplax H1 and at least an initial characterization of these cell types in the other three Placozoan species.

However, the manuscript presents a series of issues which in my opinion need to be addressed before proceeding to publication.

1) The major issue is the absence of proper controls for the FISH. Given the very small size of most Trichoplax cells and the diffuse or low-level nature of some ISH signals (see for example Ta63996 in Fig1A; TaChymotripsin in Fig1B; TaIntelectin in Fig5D; TaELPE in Fig6A1) it is difficult to understand whether the signal shown is a real one or some background noise. This problem is only partially alleviated by showing FISH on dissociated cells. While the most convincing control would be using sense probes for each of the studied mRNAs, using at least a scrambled probe might be enough.

2) A related problem is that in some cases so many different signals are shown at the same time that the pictures become extremely difficult to interpret (see for example: Fig2B; Fig4A; Fig5F; Fig6A1; Fig7A and G).

While the possibility to simultaneously show the expression of several genes is of course one of the great advantages of multiplex FISH, the different channels should also be shown separately (at least as supplementary pictures) to allow for a better appreciation of the expression patterns.

3) I do not feel qualified enough to judge the robustness and pertinence of the statistical tools exploited. Independent advice should be obtained from an expert.

4) Some of the data related to the feeding behavior of Trichoplax are not particularly convincing. In particular, in Fig8 A-D the high 'background' level of cyan stain apparently due to algal debris makes it very difficult to appreciate the increase which would be due to the release of secreted LC granules. In general, this whole paragraph does not bring much to the manuscript.

5) The manuscript contains a certain number of overstatements or not fully substantiated assertions which should be eliminated. The strongest one (already present in the title) is the assumption that the expression of transcripts bearing some similarities to arminins is sufficient to qualify the expressing cells as 'implicated in innate immunity'. This is a simple hypothesis or speculation which in the present form of the manuscript is not supported by functional evidence and should therefore be avoided.

6) The nomenclature of some transcripts is sometimes unclear and confusing. For example, three genes encoding 'Arminin-Like Peptides' (AMP1-3) are mentioned. AMP1 is Ta55945 and AMP2 is Ta56030 , but no identity is given for AMP3. On the other hand 'a third Arminin-like prepropeptide (Ta 60631)' is referred to without specifying whether this is the same as AMP3. In several other cases, the same transcripts appear to be referred to sometimes by their number, sometimes by a name (this is the case for example for Intelectin/Ta60661; OligoBP/Ta63702; Astacin/Ta26557). These ambiguities further complicate the reading of a manuscript which is already extremely dense in terms of nomenclature and should be corrected.

7) Some other inconsistencies/mistakes scattered throughout the manuscript should be corrected. Among these:

Page 8, (S2 Fig) should be (S1 Fig).

Page 10, (Fig 4D, E) should be (Fig 5D, E).

Page 14, (Fig 5E2, E3) should be (Fig 6E2, E3).

Page 14, (Fig5E4) should be (Fig 6E4).

Pages 31 and 32, the same paper (Mayorova et al., Biol. Open, 2019) is quoted twice, once as ref17, another as ref32.

8) The manuscript contains too much unsubstantiated speculations about possible homologies between the functions of some cells types in Trichoplax and those of cell types know in other (sometimes evolutionarily extremely distant) metazoans. This makes for an overlong manuscript, whose readability and intelligibility would benefit a lot from a substantial shortening.

Reviewer #2: The manuscript utilizes new cell type data (previous publications) from Placozoans to infer cell types (using mRNA in situ hybridization). Gene expression analysis coupled with behavioral observations help suggest complex secretory interactions along the "dorsal-ventral" axis.

Overall, this study is appealing to infer molecular, cellular and behavioral homology among distantly related taxonomic groups. What I felt this study failed to do is to appeal to the broad audience of a PLOS One reader. Beginning from Figure 1 through the end, the article could do a better job over conveying the message to the viewers.

Figure 1 – For the broad reaching scope of PLOS One, this figure fails to connect the audience to the animal (a photo should be added, even a snap from the Supplemental movie showing algae). I think Figure 1 could be a supplemental figure and should be summarized as a main figure. Perhaps you could join in Figure 2 and make one nice summary figure for figure 1. If you incorporated Figure 2 CD should have the probably 2 photos (if you are merging the transmitted light?). I would have one of the fluorescence only, and one with 4 overlaid channels – Or better, one fluorescent and just a transmitted photo alongside. This would make the photos not appear blown out.

Figure 3 - Beautiful photos, scale bars sometimes hard to see through out the paper.

Figure 4-7 - I like the attempt to quantify and project cell types of the ventral surface by using co-expression.

Figure 8 - Creative way to understand cell behaviors.

I think overall this paper could benefit with greater explanation (summary figure) on the localization of these patterns in a 3D model and should reflect how cell types are dictating perspective function in the animal (e.g. digestive, anti-microbial).

The authors are one of the few to perform in situ hybridization on these animals. They claim to have used commercial reagents but do not provide a protocol to recreate these experimental conditions for in situs. If they are introducing a new protocol, it would be nice to include previously published patterns as a reference. Perhaps a figure summarizing cell type data (compared to known placozoan expression patterns would be helpful to summarize) if the authors do not want to do more in situs on previously published genes. This would increase the impact of the publication.

The authors refer to these animals as irregular shaped and lacking axial symmetry (without citation). Instead their data and previous in situs clearly show axial organization in terms of a primitive digestive layer. I think their work continues to support the idea that there is symmetry (oral-aboral?) in these animals.

Overall I think restructuring the main figures into a condensed story would help improve the link to viewership for the broad audience of PLOS One. I did enjoy reading this paper. Good luck!

6. PLOS authors have the option to publish the peer review history of their article (what does this mean? ). If published, this will include your full peer review and any attached files.

**Do you want your identity to be public for this peer review?** For information about this choice, including consent withdrawal, please see our Privacy Policy .

Reviewer #1: No

Reviewer #2: No

---

## [Author Response · Author response to Decision Letter 1]

4 Dec 2024

Response to Reviewers

Manuscript PONE-D-24-38008

Dear Dr. Schubert,

Thank you for giving us the opportunity to submit a revised draft of the manuscript “Placozoan secretory cell types implicated in feeding, innate immunity and regulation of behavior” for publication in the PLOS One. We appreciate the time and effort that you and the reviewers dedicated to providing feedback on our manuscript and are grateful for the insightful comments on and valuable improvements to our paper. We have incorporated most of the suggestions made by the reviewers. Those changes are highlighted within “Revised Manuscript with Track Changes”. Please see below, in red, for a point-by-point response to the reviewers’ comments and concerns.

Following suggestions by the reviewers, we added an introductory figure (Fig 1) and summary figure (Fig 9), moved the previous Fig 1 to Supplementary Material (S1 Fig), modified existing figures (Figs 2, 6, 7) and added supplementary figures (S2 Fig, S9 Fig) to include color separated fluorescence images. Previous S1 Fig was renamed S7 Fig, previous S7 Text was renamed S8 Text, and previous S9, 10 Movies were renamed S11, 12 Movies. We changed the nomenclature referring to the four studied species/haplotypes of Placozoa, as follows: Trichoplax adhaerens H1 (TH1); Trichoplax sp. H2 (TH2); Hoilungia hongkongensis (HH13); Cladtertia collaboinventa (CH23) because our previous naming scheme implied that H23 is a member of the genus Hoilungia, which is incorrect.

We noticed two mistakes and a missing reference on pages 20 and 21 and corrected them as follows:

Original sentence. Metacells classified as “epithelial” or “lower epithelial” in scRNAseq studies of TH1 [6,31] based on expression of genes associated with the structure and function of cilia highly express genes with structural similarities to precursors of arminins, antimicrobial peptides found in Hydra magnipapillata endodermal cells [33,76].

Corrected sentence. Metacells classified as “epithelial” or “lower epithelial” in scRNAseq studies of TH1 [6,31] based on expression of genes associated with the structure and function of cilia highly express genes with structural similarities to precursors of arminins, antimicrobial peptides found in Hydra endodermal cells [33,76, 77].

Original sentence. Genetic knockdown of arminin expression impairs the ability of H. magnipapillata to reestablish a microbiome resembling its native microbiome following destruction of its microbiome by antibiotic treatment.

Corrected sentence. Genetic knockdown of arminin expression impairs the ability of H. vulgaris to reestablish a microbiome resembling its native microbiome following destruction of its microbiome by antibiotic treatment.

Reviewers' Comments to the Authors:

Reviewer #1: This paper by Mayorova and colleagues is an ambitious first attempt at correlating single cell transcriptomics data and morphological data (obtained by electron microscopy and confocal microscopy either by this or other teams) across four species belonging to phylum Placozoa. To achieve this, the authors mostly exploit a multiplex fluorescent mRNA In Situ Hybridization approach on Trichoplax adhaerens H1, and use this as a template for establishing comparison with the single cell transcriptomes from Trichoplax H2 and two Hoilungia haplotypes , H13 and H23.

Such an approach would indeed lead to a very-fine level characterization of the different cell types constituting the epithelia of Trichoplax H1 and at least an initial characterization of these cell types in the other three Placozoan species.

However, the manuscript presents a series of issues which in my opinion need to be addressed before proceeding to publication.

1) The major issue is the absence of proper controls for the FISH. Given the very small size of most Trichoplax cells and the diffuse or low-level nature of some ISH signals (see for example Ta63996 in Fig1A; TaChymotripsin in Fig1B; TaIntelectin in Fig5D; TaELPE in Fig6A1) it is difficult to understand whether the signal shown is a real one or some background noise. This problem is only partially alleviated by showing FISH on dissociated cells. While the most convincing control would be using sense probes for each of the studied mRNAs, using at least a scrambled probe might be enough.

Author Response. We previously optimized protocols for FISH with RNAscope probes (Advanced Cell Diagnostics) in Trichoplax wholemounts and dissociated cell preparations to achieve high signal and low background, and added citations to those publications [18,25] in the Methods section (page 26). We included detailed descriptions of the fixation protocols in the Methods sections of the present study (pages 26-28) because our protocols differ from protocols recommended by ACD and because we added steps for immunolabelling in experiments on dissociated cell preparations. We explain in Methods that the negative control probe gave no label and added the statement “as did the sense probes for two of the genes studied (TaYPFFamide, Cat. #488721; and Ta 26557, Cat #1222671-C3)”. The probes for genes specifically expressed in different metacells identified in single cell transcriptomes (RNAseq) [6, 31] labeled populations of cells with distinctive distributions and revealed the expected patterns of co-expression. For example: LC co-expressed Tetraspanin, 63996 and Gaba Transporter and were in the same area as LC labeled with a probe for a different LC specific gene in a study using a different FISH protocol [6]; cVEC co-expressed trypsin, chymotrypsin and sPLA2 and had the same distribution as cells labeled for chymotrypsin by a different FISH protocol [6]; cells labeled with our probe for SIFGa prepropeptide had similar distributions as cells labeled by a purified antibody against SIFGa [23] and some of them co-expressed FFNPa precursor, as predicted by RNAseq studies [6,31].

2) A related problem is that in some cases so many different signals are shown at the same time that the pictures become extremely difficult to interpret (see for example: Fig2B; Fig4A; Fig5F; Fig6A1; Fig7A and G).

While the possibility to simultaneously show the expression of several genes is of course one of the great advantages of multiplex FISH, the different channels should also be shown separately (at least as supplementary pictures) to allow for a better appreciation of the expression patterns.

Author Response. Thank you for these suggestions. We added color separated images of the xz views in Fig 2B, 6A1, B1, and 7A, E, G. We added supplemental figures showing color separated views of the images in Figs 2, 5D, E and 6F. We added text to the legend for the revised Fig. 6B (page 11) explaining that “The yellow channel in the color-merged xy and xz images were dislayed with gamma= 0.54 to enhance the visibility of the weakly labeled cells. The yellow channel is displayed with gamma=1.0 (linear) in the upper xz image”. We added text to the legend for the revised Fig. 6F explaining that “The yellow channel was displayed with gamma=0.75 to enhance the visibility of weakly stained cells. Images displayed with gamma=1.0 (linear) are shown in S9 Fig.

3) I do not feel qualified enough to judge the robustness and pertinence of the statistical tools exploited. Independent advice should be obtained from an expert.

4) Some of the data related to the feeding behavior of Trichoplax are not particularly convincing. In particular, in Fig8 A-D the high 'background' level of cyan stain apparently due to algal debris makes it very difficult to appreciate the increase which would be due to the release of secreted LC granules. In general, this whole paragraph does not bring much to the manuscript.

Author Response. In our opinion, one of the key findings of the present study was that a subset of monociliated ventral epithelial cells (cVEC) express digestive enzymes. Our experiments monitoring secretion during feeding episodes in live animals with LipidTox, a dye that stains intact LC granules, FM1-43, which stains the secreted contents of LC granules [4], and BZiPAR, an indicator of trypsin activity [40], demonstrate that both LC and cVEC participate in decomposing algae. Secretion of trypsin was evident during feeding episodes due to hydrolysis of BZiPAR and the consequent increase in green fluorescence. We used a point-scanning spectral confocal microscope and linear-unmixing algorithms for these experiments because they required monitoring fluorophores with overlapping excitation/emission spectra. The slow scan speeds made it difficult to directly visualize secretion of LipidTox-stained LC. However, our previous observations with microscopes allowing faster frame rates (16 fps) [4] showed that LC granule secretion coincided with the appearance of FM1-43 stained spots, so we used the appearance of these spots to monitor LC granule secretion in this study. We explained our rationale in the Methods section. We appreciate that the spots are hard to see due to their low abundance compared to FM1-43 stained algae and debris. We already outlined and marked FM1-43 stained spots in Fig 8b and we have now added arrowheads to S10 Fig (previously S2Fig) to mark the FM1-43 stained spots.

5) The manuscript contains a certain number of overstatements or not fully substantiated assertions which should be eliminated. The strongest one (already present in the title) is the assumption that the expression of transcripts bearing some similarities to arminins is sufficient to qualify the expressing cells as 'implicated in innate immunity'. This is a simple hypothesis or speculation which in the present form of the manuscript is not supported by functional evidence and should therefore be avoided.

Author Response. Our study shows that pVEC and a gland cell type express genes that structurally resemble the precursors of arminins, a type of antimicrobial peptide implicated in innate immunity in Hydra [33], and that DEC express multiple genes identified as intelectins based on the presence of a fibrinogen-related domain [36]. We provide an overview of the literature implicating arminins in innate immunity in Hydra (page 21) and intelectins in innate immunity in Porifera and chordates (page 23). We hypothesize that arminins and intelectins may serve similar functions in Placozoa. Many authors have made inferences about the mechanisms employed for defense in one group of animals based on expression of genes implicated in defense in other animals, for example [36]. The title of our article was intended to attract attention of scientists interested in mechanisms involved in feeding, innate immunity and signaling in Placozoa. The text and citations explain our rationale for implicating genes resembling arminins and intelectins in innate immunity. We changed a sentence in the first paragraph of the Discussion (p. 17) from “The peripheral part of the ventral epithelium and the dorsal epithelium contain secretory cells that express peptides and/or glycoproteins implicated in defense” to “The peripheral part of the ventral epithelium and the dorsal epithelium contain secretory cells that express genes with sequences resembling precursors of secretory peptides and/or glycoproteins implicated in defense in other animals.”

6) The nomenclature of some transcripts is sometimes unclear and confusing. For example, three genes encoding 'Arminin-Like Peptides' (AMP1-3) are mentioned. AMP1 is Ta55945 and AMP2 is Ta56030 , but no identity is given for AMP3. On the other hand 'a third Arminin-like prepropeptide (Ta 60631)' is referred to without specifying whether this is the same as AMP3. In several other cases, the same transcripts appear to be referred to sometimes by their number, sometimes by a name (this is the case for example for Intelectin/Ta60661; OligoBP/Ta63702; Astacin/Ta26557). These ambiguities further complicate the reading of a manuscript which is already extremely dense in terms of nomenclature and should be corrected.

Author Response. We agree that our nomenclature was at times confusing and have modified the text accordingly. We included ID numbers in the text for all studied genes, except prepropeptides, which are listed in S8 Text. We named Ta 60631 “AMP3” (page 9) and changed the name of Ta 64402 to “AMP4” (page 11). We use gene names in the figures when unambiguous, rather than ID numbers, because the names are more meaningful to readers.

7) Some other inconsistencies/mistakes scattered throughout the manuscript should be corrected. Among these:

Page 8, (S2 Fig) should be (S1 Fig). S2 Fig was correct, but now has been renamed S7 Fig.

Page 10, (Fig 4D, E) should be (Fig 5D, E). Corrected.

Page 14, (Fig 5E2, E3) should be (Fig 6E2, E3). Corrected.

Page 14, (Fig5E4) should be (Fig 6E4). Corrected.

Pages 31 and 32, the same paper (Mayorova et al., Biol. Open, 2019) is quoted twice, once as ref17, another as ref32. Corrected.

8) The manuscript contains too much unsubstantiated speculations about possible homologies between the functions of some cells types in Trichoplax and those of cell types know in other (sometimes evolutionarily extremely distant) metazoans. This makes for an overlong manuscript, whose readability and intelligibility would benefit a lot from a substantial shortening.

Author Response. We believe that pointing out the similarities and differences between cells in Placozoa and cells that perform analogous functions in other animals will be of interest to readers seeking to understand the evolution and functions of metazoan cell types and help them to appreciate the relevance of our findings to these topics. However, we agree that some parts of the Discussion were too long and detailed. We shortened the section pointing out the similarities between Trichoplax LC granules and mammalian NKC lytic granules (pages 19-20) as follows:

The outer part of the LC apical granule binds osmium, as evident by electron microscopy in thin sections from samples fixed with osmium, indicating the presence of unsaturated lipids [32]. In thin sections of LC from frozen and freeze substituted samples, the apical granules contained variable amounts of electron dense material as well as small vesicles whose content resembled cytoplasm. Granules deeper in the cell bodies of LC contained progressively less electron dense material and larger membrane enclosed vesicles as well as finger-shaped protrusions from the membrane surrounding the granule. The ultrastructure of LC apical granules bears some resemblance to that of lytic granules in rat natural killer cells (NKC), which likewise contain electron dense cores and small membrane-enclosed vesicles [65–67]. Natural killer cell granules are considered secretory lysosomes or lysosome related organelle (LRO) because they possess characteristics of both secretory granules and lysosomes. The affinity of acidophilic Lysotracker dyes for LC granules demonstrates that their content is acidic [4,16], like the content of NKC granules. Lipophil metacells highly express lysosomal proteins including lysosomal membrane associated protein 2 (LAMP2), hydrolases and multiple H-transporting V-type ATPases, leading us to conclude that LC granules may likewise represent a type of LRO.

The presence of lipids in LC granules is intriguing and not a feature shared with NKC granules. However, LRO in some other types of cells do contain lipids [68,69]. Lipophil metacells highly express several fatty-acid binding proteins [6] that might serve as chaperones to deliver lipids to their granules. Trichoplax can lyse and consume cyanobacteria in addition to microalgae [4], indicating that the content of LC granules can penetrate both bacterial and eukaryotic membranes. The identities of the lytic components and their molecular targets in microalgae and cyanobacteria are important questions that remain to be addressed.

Reviewer #2: The manuscript utilizes new cell type data (previous publications) from Placozoans to infer cell types (using mRNA in situ hybridization). Gene expression analysis coupled with behavioral observations help suggest complex secretory interactions along the "dorsal-ventral" axis.

Overall, this study is appealing to infer molecular, cellular and behavioral ho

---

## [Decision Letter · Decision Letter 1]

8 Jan 2025

PONE-D-24-38008R1Placozoan secretory cell types implicated in feeding, innate immunity and regulation of behaviorPLOS ONE

Dear Dr. Smith,

Thank you for submitting your manuscript to PLOS ONE. After careful consideration, we feel that it has merit but does not fully meet PLOS ONE’s publication criteria as it currently stands. Therefore, we invite you to submit a revised version of the manuscript that addresses the points raised during the review process.

We look forward to receiving your revised manuscript.

Kind regards,

Michael Schubert

Academic Editor

PLOS ONE

Journal Requirements:

Reviewers' comments:

Reviewer's Responses to Questions

**Comments to the Author**

1. If the authors have adequately addressed your comments raised in a previous round of review and you feel that this manuscript is now acceptable for publication, you may indicate that here to bypass the “Comments to the Author” section, enter your conflict of interest statement in the “Confidential to Editor” section, and submit your "Accept" recommendation.

Reviewer #1: (No Response)

Reviewer #2: All comments have been addressed

2. Is the manuscript technically sound, and do the data support the conclusions?

Reviewer #1: Partly

Reviewer #2: Yes

3. Has the statistical analysis been performed appropriately and rigorously? 

Reviewer #1: Yes

Reviewer #2: I Don't Know

4. Have the authors made all data underlying the findings in their manuscript fully available?

Reviewer #1: Yes

Reviewer #2: Yes

5. Is the manuscript presented in an intelligible fashion and written in standard English?

Reviewer #1: Yes

Reviewer #2: Yes

6. Review Comments to the Author

Reviewer #1: The addition of separated channel images as supplementary figures has indeed improved the interpretation of the multiplex FISH data.

Additional images, such as Fig1, do make the manuscript more appealing to a wider audience of non Placozoan specialists (although Fig1C should be more appropriately described as 'A drawing of main cell types in TH1 as observed by Transmitted Electron Microscopy').

The gene nomenclature has been harmonized throughout the manuscript, making it much easier to read.

Mistakes and typos have been mostly corrected (although 'Rhodamonas' should be corrected to 'Rhodomonas').

The authors chose to keep lots of data and discussion which in my opinion are not strictly pertinent or excessively speculative. This is a legitimate choice but unfortunately it distracts the readers' attention away from the most interesting and remarkable achievement of this work, i.e. the generation of a detailed, single-cell level atlas of gene expression patterns in a little-studied and not easily amenable metazoan taxon. Personally, I still believe that a more concise and matter-of-fact version of the manuscript would be preferable and have a much stronger and wider impact. However, I agree that my should be balanced with that of reviewer n°2 and with the editor's advice.

Reviewer #2: I think the authors greatly improved my main criticism, which was that they needed to do a better job connecting with a broader audience. They improved that with the addition of Figure 1 and by creating the summary figures as I recommended. My only point that was not addressed was comparing existing expression pattern data (like Trox-2) or other markers to help explain the domains of patterning, but I understand if this extends away from the focus of the paper. Overall I think this work would be a nice fit for PLOS One.

7. PLOS authors have the option to publish the peer review history of their article (what does this mean? ). If published, this will include your full peer review and any attached files.

**Do you want your identity to be public for this peer review?** For information about this choice, including consent withdrawal, please see our Privacy Policy .

Reviewer #1: No

Reviewer #2: No

---

## [Author Response · Author response to Decision Letter 2]

16 Jan 2025

Reviewers' Comments to the Authors:

Reviewer #1: The addition of separated channel images as supplementary figures has indeed improved the interpretation of the multiplex FISH data.

Additional images, such as Fig1, do make the manuscript more appealing to a wider audience of non Placozoan specialists (although Fig1C should be more appropriately described as 'A drawing of main cell types in TH1 as observed by Transmitted Electron Microscopy').

Author; We modified legend for Fig1C as suggested.

The gene nomenclature has been harmonized throughout the manuscript, making it much easier to read.

Mistakes and typos have been mostly corrected (although 'Rhodamonas' should be corrected to 'Rhodomonas').

Author: Corrected.

The authors chose to keep lots of data and discussion which in my opinion are not strictly pertinent or excessively speculative. This is a legitimate choice but unfortunately it distracts the readers' attention away from the most interesting and remarkable achievement of this work, i.e. the generation of a detailed, single-cell level atlas of gene expression patterns in a little-studied and not easily amenable metazoan taxon. Personally, I still believe that a more concise and matter-of-fact version of the manuscript would be preferable and have a much stronger and wider impact. However, I agree that my should be balanced with that of reviewer n°2 and with the editor's advice.

Author: We modified the Introduction and Discussion for clarity and to make them more concise.

Line numbers refer to “Revised Manuscript with Track Changes”. Note that the reference numbering in this document is incorrect because the reference list includes deleted references.

Introduction

Modified text at lines 79-80, 138, 159-162.

Deleted text at lines 101, 105, 116, 131-133, 137-139, 144, 162.

Discussion

Modified text at lines 747-749, 830-835, 886, 905, 944-945, 984, 1014-1015, 1113-1122.

Deleted text at lines 764, 806, 835, 850, 881, 883, 946, 984, 986, 1014, 1017, 1122.

Conclusions section of the Discussion was modified for clarity and to better explain the relevance of our work to efforts to understand the evolution of secretory cells (lines 1125-1176).

Reviewer #2: I think the authors greatly improved my main criticism, which was that they needed to do a better job connecting with a broader audience. They improved that with the addition of Figure 1 and by creating the summary figures as I recommended. My only point that was not addressed was comparing existing expression pattern data (like Trox-2) or other markers to help explain the domains of patterning, but I understand if this extends away from the focus of the paper. Overall I think this work would be a nice fit for PLOS One.

---

## [Editor Report · Decision Letter 2]

19 Jan 2025

Placozoan secretory cell types implicated in feeding, innate immunity and regulation of behavior

PONE-D-24-38008R2

Dear Dr. Smith,

We’re pleased to inform you that your manuscript has been judged scientifically suitable for publication and will be formally accepted for publication once it meets all outstanding technical requirements.

Kind regards,

Michael Schubert

Academic Editor

PLOS ONE

---

## [Editor Report · Acceptance letter]

PONE-D-24-38008R2

PLOS ONE

Dear Dr. Smith,

I'm pleased to inform you that your manuscript has been deemed suitable for publication in PLOS ONE. Congratulations! Your manuscript is now being handed over to our production team.

Kind regards,

on behalf of

Dr. Michael Schubert

Academic Editor

PLOS ONE